# Visuo-frontal interactions during social learning in freely moving macaques

Melissa Franch[1], Sudha Yellapantula[2], Arun Parajuli[1], Natasha Kharas[1], Anthony Wright[1], Behnaam Aazhang[2] & Valentin Dragoi[1,2,3,4 ✉]

Social interactions represent a ubiquitous aspect of our everyday life that we acquire by interpreting and responding to visual cues from conspecifics[1]. However, despite the general acceptance of this view, how visual information is used to guide the decision to cooperate is unknown. Here, we wirelessly recorded the spiking activity of populations of neurons in the visual and prefrontal cortex in conjunction with wireless recordings of oculomotor events while freely moving macaques engaged in social cooperation. As animals learned to cooperate, visual and executive areas refined the representation of social variables, such as the conspecific or reward, by distributing socially relevant information among neurons in each area. Decoding population activity showed that viewing social cues influences the decision to cooperate. Learning social events increased coordinated spiking between visual and prefrontal cortical neurons, which was associated with improved accuracy of neural populations to encode social cues and the decision to cooperate. These results indicate that the visual-frontal cortical network prioritizes relevant sensory information to facilitate learning social interactions while freely moving macaques interact in a naturalistic environment.

Social cooperation is a complex behaviour whereby animals look at each other to perceive and interpret social cues to decide whether to interact[2,3]. These cues range from body language and facial expressions to rewarding stimuli and have been notoriously difficult to identify and analyse so far. Although previous studies have been instrumental in our understanding of the neural encoding of specific social variables, such as reward value[4,5], actions[6–8], agent identity[9,10] and social rank[11–13], they did not attempt to examine the neural processes that mediate the emergence of visually guided social decision-making and cooperation behaviour. Indeed, traditional studies examining the neural underpinnings of social behaviour have typically been completed using stationary animals performing passive tasks using synthetic stimuli. Technical limitations have prevented the recording of visual cues to further examine how they are used to shape social behaviour while animals interact with each other.

Macaques exhibit social behaviour in natural and laboratory environments, such as cooperation[14–16] and competition[17], and they strategically acquire social information from viewing eye and facial expressions[3], hence making them an ideal model to study social cognition. Previous work in non-human primates has identified brain regions that are activated when viewing other agents in person or socially interacting animals in images and videos[18–20], but examining how the brain processes social signals originating from interacting conspecifics in real time to initiate goal-directed behaviour has not, to our knowledge, been explored until now. Here, we developed an approach that combines behavioural and wireless eye tracking and neural monitoring to study how pairs of freely moving, interacting macaques use visually guided signals to learn social cooperation for food reward. Our approach allows us to investigate behaviours and neural computations promoting cooperation and how they change over time while learning to cooperate. Harnessing the versatility of wireless neural and eye-tracking recordings combined with markerless behavioural tracking[21], we examined how pairs of macaques learn to cooperate by identifying the visual cues used to guide decision-making along a visual-frontal cortical circuit.

Two unique and familiar pairs of macaques learned to cooperate for food reward across weeks. Owing to macaques' natural social hierarchy, each pair consisted of a subordinate monkey ('self') and a dominant monkey ('partner'). Animals cooperated in an arena, separated by a clear divider, so they could visually but not physically interact. Each monkey could freely move around their side, and each monkey had their own push button. At the start of a trial, perceivable but remote pellets dispensed into the animals' respective trays, and animals could cooperate at any time by simultaneously pushing and holding individual buttons that moved their trays, delivering reward to each animal (Fig. 1a and Methods). A trial began when pellets dispensed and ended when the trays reached the animals (each session included 100–130 cooperation trials: 18 learning sessions with monkey pair 1 and 17 sessions with monkey pair 2). Button pressing was recorded for each monkey while neural and eye-tracking data were wirelessly recorded simultaneously from the subordinate ('self') monkey (Fig. 1b,c). We chronically recorded from populations of neurons in the midlevel visual cortex (area V4) and dorsolateral prefrontal cortex (area dlPFC) of the several 'self' animals, as these are key areas involved in processing complex visual features[22–26] and planning social actions[6,27,28]. In each monkey (n = 2), we used dual

[1]Deparment of Neurobiology and Anatomy, McGovern Medical School, University of Texas, Houston, TX, USA. [2]Department of Electrical and Computer Engineering, Rice University, Houston, TX, USA. [3]Neuroengineering Initiative, Rice University, Houston, TX, USA. [4]Houston Methodist Research Institute, Houston, TX, USA. ✉e-mail: vdragoi@rice.edu

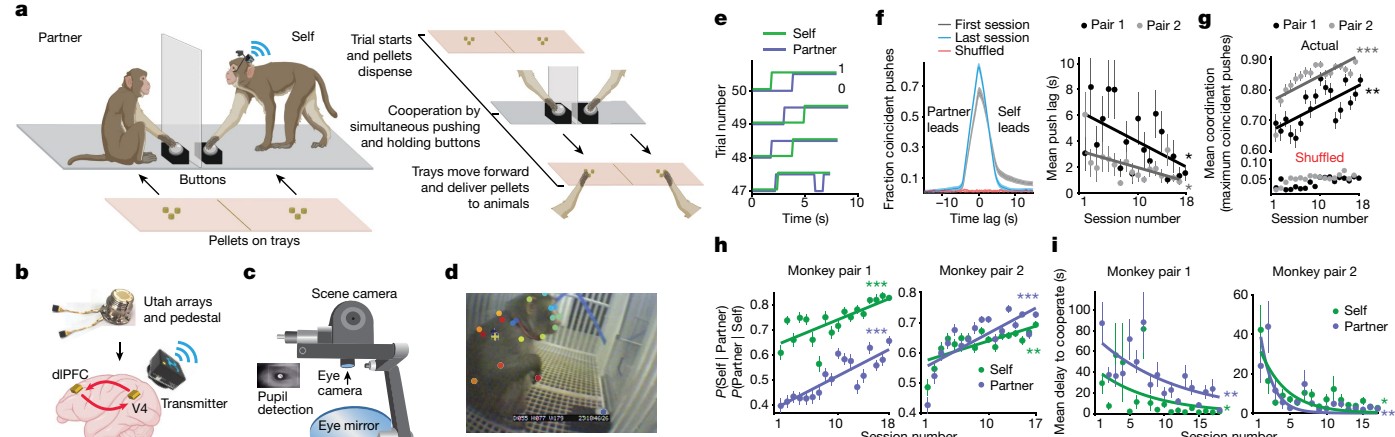

**Fig. 1 | Tracking of behavioural, oculomotor and neural events during learning cooperation. a**, Behavioural task. Two animals learned to cooperate for food reward. Left, cooperation paradigm. Right, trial structure. **b**, Wireless neural recording equipment (Blackrock Neurotech). Red arrows represent information processing between areas. **c**, Wireless eye tracker and components. **d**, DeepLabCut labelling of partner-monkey and buttons from the eye tracker's scene camera. The yellow cross represents self-monkey's point of gaze. **e**, Example voltage traces of each animal's button-push activity from pair 1. A line increase to 1 indicates the monkey began pushing. **f**, Left, example CCGs of pair 1's button pushes from the first and last session, using actual and shuffled data. Self-monkey leads cooperation more often in early sessions, as the peak occurs at positive time lag (2 s). Right, session average time lag between pushes

when maximum coincident pushes occur. Pair 1: $P = 0.03$ and $r = -0.5$; pair 2: $P = 0.02$ and $r = -0.5$. **g**, Push coordination. Session average maximum number of coincident pushes (that is, peaks from CCGs). Pair 1: $P = 0.001$ and $r = 0.7$; pair 2 $P = 0.008$ and $r = 0.7$. **h**, Session average conditional probability to cooperate for each monkey. Pair 1: $P = 0.0004$, $r = 0.7$ and $P = 6.02 \times 10^{-6}$, $r = 0.8$; pair 2: $P = 0.001$, $r = 0.7$ and $P = 0.0004$ and $r = 0.8$, self and partner, respectively. **i**, Session average delay to cooperate or response time for each monkey. Pair 1: $P = 0.01$, $r = -0.6$ and $P = 0.001$, $r = -0.6$; pair 2: $P = 0.01$, $r = -0.6$ and $P = 0.006$, $r = -0.6$, self and partner, respectively. All $P$ values are from linear regression, and $r$ is Pearson correlation coefficient. On all plots, circles represent the mean, with error bars s.e.m. *$P < 0.05$, **$P < 0.01$, ***$P < 0.001$. Illustrations in **a** and **b** were created using BioRender.

Utah arrays to stably record from the same neural population (averaging 136 units per session in monkey 1 (M1) and 150 units per session in monkey 2 (M2), including single units and multi-units; M1, 34 V4 cells and 102 dlPFC cells; M2, 104 V4 cells and 46 dlPFC cells) across several sessions[29] (Figs. 1b and 3a and Extended Data Fig. 3). Videos from the wireless eye tracker's scene camera capturing the self-monkey's field of view were analysed with DeepLabCut[21] to identify objects of interest that the animal viewed in their environment (Fig. 1d and Methods).

## Learning social cooperation

To quantify changes in cooperation performance over time, we analysed features of both animals' actions, such as the coordination of their push onset and duration, the conditional probability of cooperating and the delay to cooperate. In a pair, each monkey's choice to cooperate can be represented as an array of zeros (not pushing) and ones (pushing, Fig. 1e). Cross-correlation analysis between the button pushes of each monkey (Methods) in each session showed that their actions are coordinated and not random (Fig. 1f, left; shuffling push times resulted in near zero coincident pushing, red plot). In the first session, for each animal pair, cross-correlograms (CCGs) peaked at 0.6 coincidences: that is, animals pushed together for 60% of the session and increased to 80–90% coincident pushing in the last session (Fig. 1f, left). Indeed, animals learned to cooperate by significantly reducing the amount of time between each of their pushes (Fig. 1f, right; all $P < 0.05$, linear regression), thereby improving response coordination across sessions (Fig. 1f,g; all $P < 0.01$, linear regression). Additionally, for each monkey, we computed the probability to cooperate given that the other monkey is pushing. Conditional probability exhibited a mean 54% increase across sessions, thus reflecting learning cooperation (Fig. 1h; $P < 0.001$, linear regression). Finally, the delay to cooperate or amount of time it takes for a monkey to respond from the trial start decreased by 93% across sessions, indicating that animals' motivation to cooperate increased during learning (Fig. 1i; $P < 0.05$, linear regression). Overall, this demonstrates that animals learned to cooperate

across sessions by improving their action coordination, conditional probability and reaction times.

## Social cues drive learning cooperation

We further examined whether animals' viewing behaviour changes during learning to cooperate by identifying the self-monkey's fixations on objects in the environment (Fig. 2a and Methods). To determine which objects were salient for cooperation, we computed the fixation rate on each object during the cooperation trial and during the intertrial interval. Fixation rates on the food reward system (pellets in tray and pellet dispenser) and partner-monkey were significantly higher during the trial, particularly before cooperation, when both monkeys began pushing, than during the intertrial period (Fig. 2b; $P < 0.01$, Wilcoxon signed-rank test). Therefore, these relevant fixations, 'view reward' and 'view partner', constitute social cues. 'View reward' includes the self-monkey's fixations on the pellets in their tray and pellet dispenser, and 'view partner' includes fixations on the partner-monkey, on both the head and body. Eye-movement analysis showed behavioural patterns in which at the beginning of a trial, the monkey typically views the reward followed by a push, while frequently looking at their conspecific before the partner's push (Fig. 2c). The ability to view the other monkey is important for cooperation, as control experiments using an opaque divider that obstructed animal's ability to view each other yielded a significant decrease in cooperation performance (Extended Data Fig. 6a).

To examine whether the relationship between social cues ('view reward' and 'view partner') and actions changes during learning, we used a Markov model to compute the probability of transitioning from one social event, or state, to another. Notably, the transitional probabilities between visually driven event pairs, but not action-driven ones, significantly increased across sessions while learning to cooperate (Fig. 2d; all $P < 0.01$, linear regression). Although each monkey pair exhibited unique transitional probability matrices for social events, this was consistent across animal pairs (Extended Data Fig. 2).

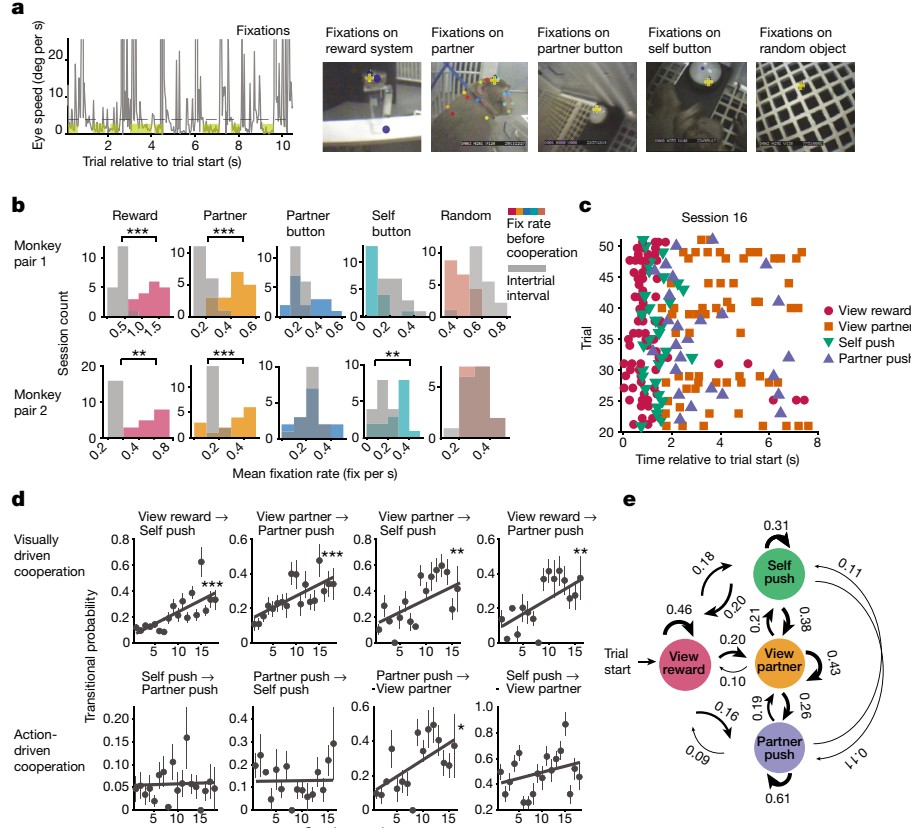

**Fig. 2 | Interactions between action and viewing while learning to cooperate. a**, Identifying fixations on various objects. Left, during fixations (highlighted in yellow), eye speed remained below threshold (dashed line) for at least 100 ms. Right, scene camera images of objects the animal viewed, labelled with DeepLabCut (coloured dots). The yellow cross represents self-monkey's point of gaze. **b**, Histograms of session mean fixation rates for each object computed during the trial (before cooperation) and intertrial interval. Asterisks represent significance of Wilcoxon signed-rank test only where fixations rates were higher during cooperation compared to intertrial period. Pair 1: $P = 0.0002, 0.0002, 0.13, 0.002$ and $0.0002$; pair 2: $P = 0.005, 0.0004, 0.95, 0.001$ and $0.7$ for fixation rates on objects listed left to right. **c**, Sequence

of action and viewing events occurring during cooperation across a random subset of trials in a session. **d**, Markov model transitional probabilities for example event pairs that begin with a viewing (top row) or action event (bottom row). Top row: $P = 0.0008, P = 0.0008, P = 0.003, P = 0.003$ and all $r = 0.7$; bottom row: $P = 0.84, 0.9, 0.01$ and $0.2$; $r = 0.6$ ('partner-push' to 'view partner'); from left to right, linear regression and Pearson correlation. Two plots on each row came from each monkey pair. Mean transitional probability with s.e.m. is plotted. For complete transitional probability matrices for each monkey, see Extended Data Fig. 2. **e**, Hidden Markov model transitional probabilities averaged across both monkey pairs for all event pairs. $*P < 0.05$, $**P < 0.01$, $***P < 0.001$.

The lowest transitional probabilities (0.1) occurred between two actions ('self-push' to 'partner-push'), indicating that fixations typically occur between pushes, and monkeys cannot simply push their button to motivate the other monkey to cooperate (Fig. 2d,e). Instead, there were high transitional probabilities (0.6–0.9) for event pairs in which fixations on a social cue occurred before or after a push (that is, 'view partner' → 'self-push' or 'self-push' → 'view partner'), thus demonstrating the importance of viewing social cues to promote cooperation (Fig. 2d,e). Indeed, we found a mean 220% increase in transitional probabilities for event pairs beginning with viewing social cues (Fig. 2d, top row; all $P < 0.01$). These analyses show that across sessions, animals become more likely to cooperate after viewing social cues, indicating that fixations on social cues drive cooperation during learning.

## Single cells respond to social variables

We investigated the relationship between neural signals and social events leading to cooperation by analysing the neurons' responses between the start of a trial and cooperation onset when both animals began pushing. We identified fixations on social cues (that is, 'view 'reward' and 'view partner') and non-social objects (that is, monkey's buttons or arena floor) within neurons' receptive fields (Fig. 3b and

Methods). Neurons in both cortical areas significantly increased mean firing rates in response to fixations on social cues compared to baseline measured during the intertrial interval (Fig. 3d,e; $P < 0.01$, Wilcoxon signed-rank test with false discovery rate (FDR) correction). A distinct feature of the 'social brain' is the ability to process information about one's self and others[6,30,31]. We explored this feature by identifying self and partner-monkey pushes or decisions to cooperate that occurred separately in time (more than 1 s from each other; Extended Data Fig. 4). Importantly, we hypothesized that the self-monkey's neurons process allocentric information during partner-choice, because the monkey views the partner during most of the partner's pushes but not during their own push (Fig. 3c). Over time, the self-monkey viewed social cues (reward or partner) before pushing, indicating that viewing social cues informs decision-making as learning emerges (Fig. 3c, bottom; $P < 0.05$, linear regression). Indeed, 70% of dlPFC cells increased their firing rate during each animal's push relative to baseline, with responses beginning 1,000 ms before push onset (Fig. 3d,e; $P < 0.01$, Wilcoxon signed-rank test with FDR correction). Notably, most dlPFC cells responded to both self and partner-choice, as opposed to just one or the other (Extended Data Fig. 4b). Overall, dlPFC neurons responded to both fixations and choice (Fig. 3e, left; all $P < 0.01$, Wilcoxon signed-rank test with FDR correction), with 55% of dlPFC neurons exhibiting mixed selectivity

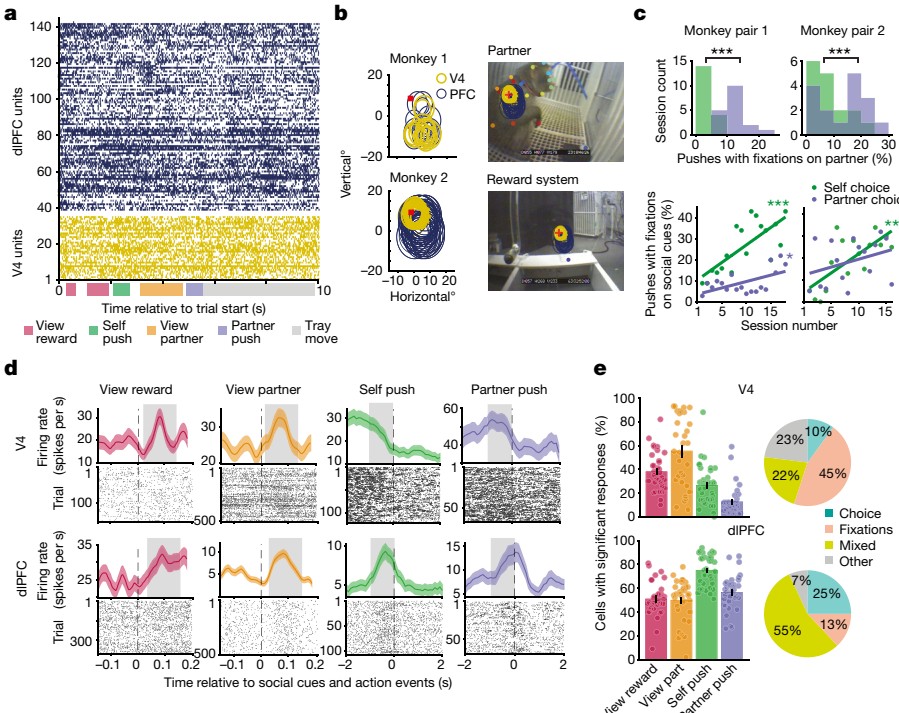

**Fig. 3 | V4 and dlPFC cell responses to social events. a**, Raster plot of spiking activity from M1's V4 units 1–35 and dlPFC units 36–140 during one trial. **b**, Social cues within neurons' receptive fields. Left, overlapping receptive fields of V4 and dlPFC neurons. V4 receptive field sizes, 4–6°; dlPFC receptive field sizes, 6–13°. The red square represents the point of fixation. Right, scene camera images measuring 35 × 28° (length × height), for which social cues were within receptive fields during fixation. **c**, Self- and partner-choice to cooperate. Top, percentage of pushes for which fixations on the partner occurred within 1,000 ms of choice in each session. Pair 1, $P = 1.91 \times 10^{-4}$; pair 2, $P = 2.44 \times 10^{-4}$; Wilcoxon signed-rank test. Bottom, percentage of pushes for which fixations on the partner and/or reward system occurred within 1,000 ms of choice in each session. Pair 1, $P = 0.0004$, $r = 0.7$ and $P = 0.03$, $r = 0.5$; pair 2, $P = 0.002$, $r = 0.7$ and $P = 0.2$, $r = 0.3$; self and partner-choice, respectively; linear regression and Pearson correlation. **d**, Peri-event time histogram and raster examples of four distinct V4 and dlPFC cells responding to each social event. Dashed lines represent event onset, and the grey shaded box represents the response period used in analyses. **e**, Significant responses. Left, percentage of cells of the total recorded (M1, 34 V4 cells and 102 dlPFC cells; M2, 104 V4 cells and 46 dlPFC cells) that exhibited a significant change in firing rate from baseline (intertrial period) during social events, averaged across sessions and monkeys. For each cell, $P < 0.01$, Wilcoxon signed-rank test with FDR correction. Right, percentage of neurons of the total recorded that responded only to choice (self and/or partner), only to fixations (reward and/or partner), both fixations and choice ('mixed') or none at all ('other'). *$P < 0.05$, **$P < 0.01$, ***$P < 0.001$.

(Fig. 3e, right). Although a fraction of V4 neurons (28%) exhibited a change in firing rate around push time, most responded to fixations on social cues (36% and 52% fixations on reward and partner, respectively). That is, in contrast to dlPFC, most cells in V4 responded only to fixations on social cues; 22% of V4 neurons exhibited mixed selectivity. These results indicate that mixed-selectivity neurons, especially in dlPFC, may support the behavioural diversity observed in our naturalistic and cognitively demanding social environment[32].

## Social learning improves the neural code

Next, we examined the ability of neural populations to encode social cues and animals' choice to cooperate during learning. A support vector machine (SVM) classifier with tenfold cross-validation was trained to decode fixations on social cues and cooperation choices from single observations (Methods). Fixations on the reward system and partner-monkey were accurately decoded from the population response in each brain area. The accuracy of decoding between the available social cues in each area increased on average by 328% during learning (Fig. 4a; all $P < 0.01$, linear regression). (Our main population results in Figs. 4 and 5 also hold when only the stable units in both areas are included in the analysis (Extended Data Fig. 7)). By contrast, whereas non-social objects, such as fixations on the self-monkey's button and random floor objects, could be reliably decoded from V4 and dlPFC activity, decoder performance did not improve across sessions

(Fig. 4b). Thus, during learning cooperation, both V4 and dlPFC selectively improve the encoding of visual features of social objects (for example, reward system and partner) but not that of other objects. Furthermore, dlPFC neurons accurately discriminated between social and non-social object categories, as decoder performance significantly improved by 228% during learning (Fig. 4c; all $P < 0.01$, linear regression). Thus, the representations of social cues and their distinction from non-social cues in dlPFC improves while animals learn to cooperate (Fig. 4c). Importantly, neural activity during visual fixations in both areas was only minimally influenced by animals' head and body movements[33,34] (Extended Data Fig. 5c,d).

We further examined whether neural populations encode each monkey's decision to cooperate. Control experiments using 'solo' and 'social' trials (Methods) showed that prepush activity is influenced by social context (Extended Data Fig. 6b–d): that is, animals' actions during learning-cooperation sessions reflect a social choice to cooperate. By decoding each animal's push from population activity, we found that in V4, choice events can be decoded only in a small number of sessions (Fig. 4d). By contrast, decoder performance in dlPFC increased on average by 5,481% while animals learned to cooperate (Fig. 4d; all $P < 0.01$, linear regression). Because we found that the self-monkey viewed different social cues during self- and partner-pushes (Extended Data Fig. 4c), we examined whether neural activity during pushing reflects decision-making or changes in visual input. Thus, we decoded each animal's choice to cooperate during two scenarios: (1) pushes with

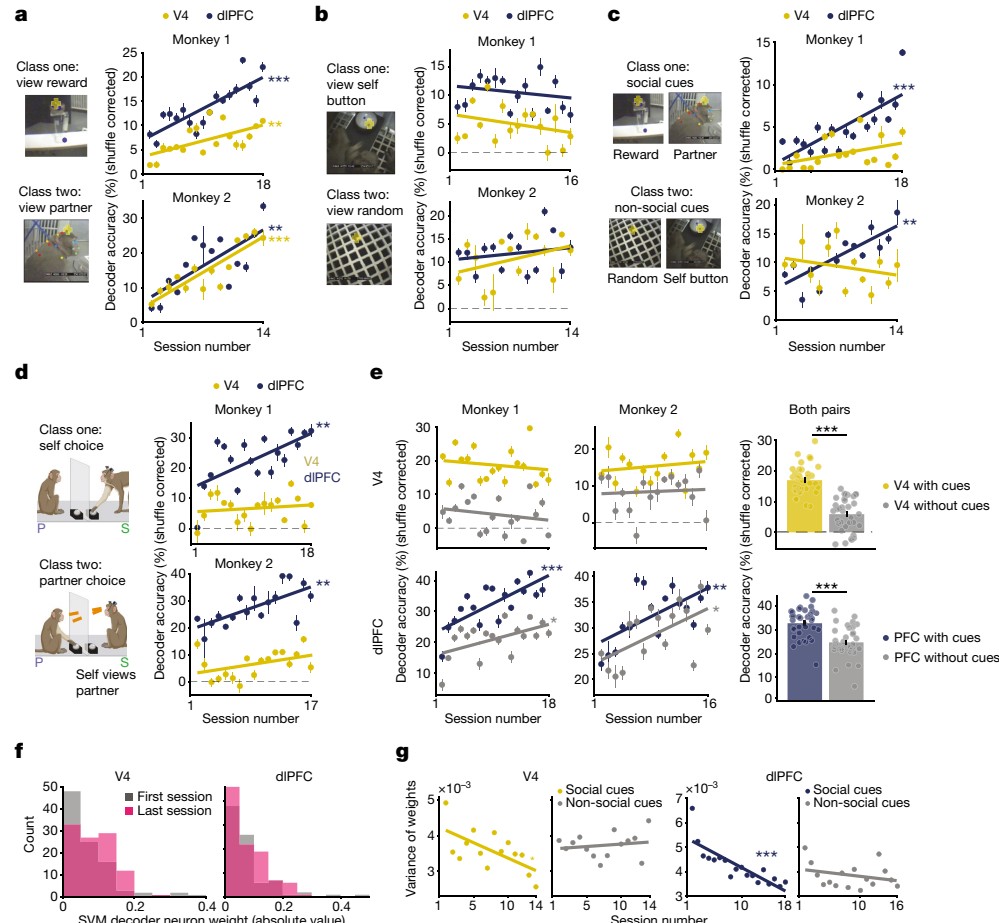

**Fig. 4 | Population encoding of social events. a**, Decoding accuracy for fixations on reward system and partner-monkey. Chance is 50%, or 0% shuffle-corrected (dashed lines). Plots display shuffle-corrected mean prediction accuracy on test observations (±s.e.m.). M1, $P = 0.006$, $r = 0.6$ and $2.31 \times 10^{-5}$, $r = 0.8$; M2, $P = 3.01 \times 10^{-5}$, $r = 0.9$ and $P = 0.004$, $r = 0.7$, V4 and dlPFC, respectively. All $P$ values in **a–e** and **g** are from linear regression; $r$ is Pearson correlation coefficient. **b**, Decoding performance for fixations on two non-social objects. M1, $P = 0.26$ and $0.41$; M2, $P = 0.18$ and $0.52$, V4 and dlPFC, respectively. **c**, Decoding performance for object categories: fixations on social and non-social cues. M1, $P = 0.08$ and $P = 0.0001$, $r = 0.8$; M2, $P = 0.3$ and $P = 0.001$, $r = 0.8$, V4 and dlPFC, respectively. **d**, Decoding performance for each animal's choice to cooperate. M1, $P = 0.54$ and $P = 0.003$, $r = 0.7$; M2, $P = 0.1$ and $P = 0.002$, $r = 0.7$, V4 and dlPFC, respectively. **e**, Viewing social cues improves choice encoding. Left, Decoding performance for each animal's choice using

pushes with preceding fixations on either social cue within 1,000 ms of push (navy and gold) compared to pushes without fixations on social cues (grey). V4 M1, $P = 0.48$ and $P = 0.4$; M2, $P = 0.49$ and $P = 0.71$. dlPFC M1, $P = 0.0002$, $r = 0.8$ and $P = 0.02$, $r = 0.5$; M2, $P = 0.008$, $r = 0.6$ and $P = 0.02$, $r = 0.6$; with and without social cues, respectively. Right, decoding accuracy for choice averaged across both monkeys for each condition. V4 $P = 7.44 \times 10^{-7}$ and dlPFC $P = 3.46 \times 10^{-6}$; Wilcoxon signed-rank test. **f**, Distribution of absolute valued neurons' weights from the SVM model for decoding social cues (V4, 98 and 102 neurons; dlPFC, 82 and 101 neurons in first and last session). **g**, Variance of weight distributions for each session from SVM models decoding social and non-social cues, as seen in **a**,**b**. V4 data are M2 (average 104 cells per session), and PFC data are M1 (average 102 cells per session). V4, $P = 0.01$, $r = -0.63$ and $P = 0.57$; dlPFC, $P = 1.67 \times 10^{-5}$, $r = -0.83$ and $P = 0.29$; social and non-social cue models, respectively. *$P < 0.05$, **$P < 0.01$, ***$P < 0.001$.

preceding fixations (within 1,000 ms) on social cues (reward system or partner) and (2) pushes without preceding fixations on social cues (the number of events in the two scenarios were balanced; Methods). In both V4 and dlPFC, decoder performance for choice was significantly reduced (by 65% and 24%, respectively) when pushes with preceding fixations on social cues were excluded (Fig. 4e, grey; $P < 0.001$, Wilcoxon signed-rank test). However, in dlPFC, the accuracy of decoding the choice to cooperate remained correlated with learning (Fig. 4e, bottom row; all $P < 0.05$, linear regression). This demonstrates that dlPFC encodes each animal's decision to cooperate and that viewing social cues during decision-making improves choice encoding. By contrast, V4 decoder performance was close to chance when button pushes preceded by fixations on social cues were removed from the analysis, indicating that V4 activity before choice mostly represents viewing social cues, not decision-making (Fig. 4e). Altogether, this demonstrates that improved encoding of egocentric (self) and allocentric (partner) choice in dlPFC, but not V4, correlates with learning

cooperation. Finally, task-irrelevant variables such as head, body and eye movement and pupil size during pushes may influence neuronal activity[34,35]. However, consistent with our previous work[36], we found only a small percentage of neurons whose activity was correlated with movements or pupil size during pushing (less than 12% of correlated neurons, $n = 1,157$ cells; Pearson correlation $P < 0.01$; Extended Data Figs. 4d,e and 5a,b).

How does the contribution of each neuron to decoder accuracy change during learning? In linear SVM models, each cell is associated with a weight representing its contribution to the decision boundary for separating one decoded event from another (that is, self and partner-choice or reward and partner stimuli). We confirmed in our models that in a session, the higher the weight magnitude, the greater the neuron's contribution to event classification (Extended Data Fig. 8c,d) and selectivity to social events (Extended Data Fig. 9c). To compare across sessions, we normalized the weights in a session and analysed the absolute value of weights of individual stable neurons

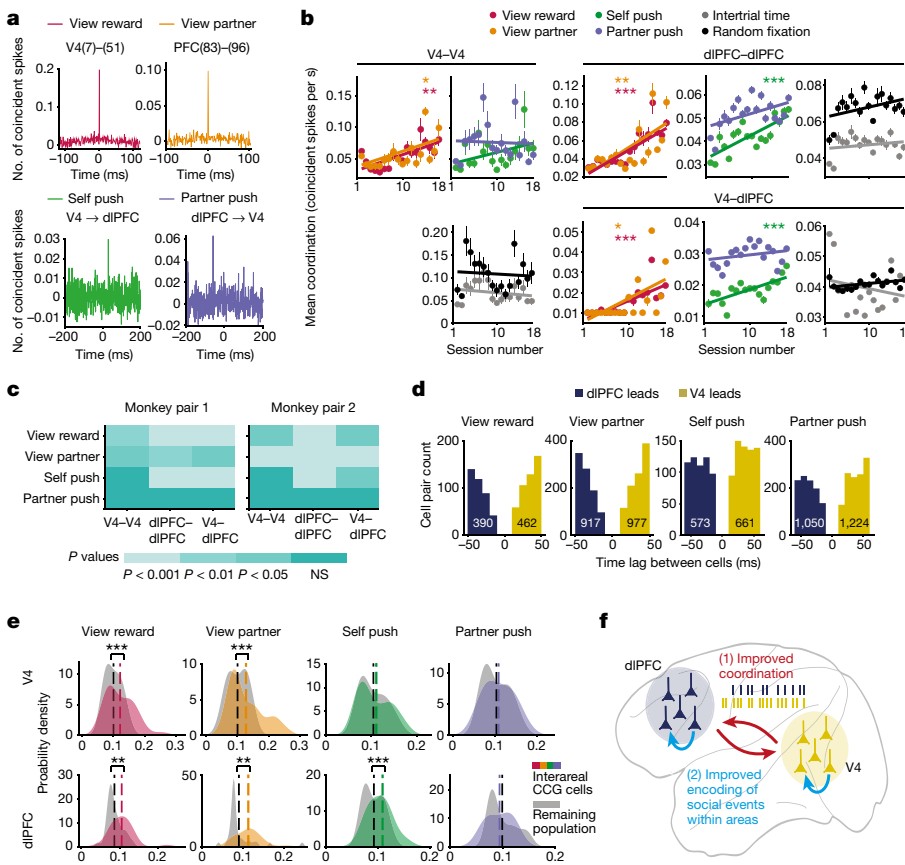

**Fig. 5 | Spike-timing coordination while learning social interactions. a**, Top, example CCGs of a V4 cell pair and a dlPFC cell pair during two social events, averaged across observations. Bottom, example CCGs of V4–dlPFC cell pairs. **b**, Temporal coordination for social and non-social events. Top, mean coordination plotted across sessions for each social event in V4, dlPFC and between areas (V4, $P = 0.001$, $r = 0.7$; $P = 0.01$, $r = 0.6$; $P = 0.09$ and $P = 0.8$. dlPFC, $P = 5.68 \times 10^{-6}$, $r = 0.9$; $P = 0.003$, $r = 0.7$; $P = 1.25 \times 10^{-4}$, $r = 0.7$ and $P = 0.07$. V4–dlPFC, $P = 2.89 \times 10^{-4}$, $r = 0.8$; $P = 0.01$, $r = 0.6$; $P = 9.93 \times 10^{-4}$, $r = 0.7$ and $P = 0.27$. Linear regression with Pearson's correlation coefficient). Bottom: mean coordination during fixations on random objects and during random events (intertrial period). V4, $P = 0.4$ and $0.7$; dlPFC, $P = 0.4$ and $0.09$; V4–dlPFC: $P = 0.4$ and $0.1$; random event and fixations, respectively. All data from M1; for M2, see Extended Data Fig. 10a. **c**, Colour map of within/between area $P$ values from

linear regression of mean coordination for each social event in each monkey. Temporal coordination increases during learning. **d**, Histograms of time-lag values of CCG peaks between all significantly correlated V4–dlPFC cell pairs across sessions and monkeys for each social event. **e**, Correlated V4–dlPFC neurons contribute more to encoding of social events. Probability density plots of decoder weights of V4 and dlPFC neurons significantly correlated and the remaining uncorrelated population during each social event. Weights were averaged across neurons in each session for each monkey and then combined. V4 from left to right, $P = 6.48 \times 10^{-4}$, $P = 6.38 \times 10^{-4}$, $P = 0.33$, $P = 0.24$; PFC: $P = 0.002$, $P = 0.001$, $P = 7.41 \times 10^{-4}$, $P = 0.14$; Wilcoxon signed-rank test. **f**, Cartoon of social learning model: increased interarea spike-timing coordination improves the encoding of social variables to mediate learning social interaction. NS, not significant.

and neural population distributions across sessions from each decoding model[37] (Methods). Surprisingly, although some (approximately 15%) individual neurons exhibited systematic changes in decoding weight or discriminability during learning (Extended Data Fig. 9d), we found more significant changes at the population level. Specifically, the variance, kurtosis and skewedness of population weight distributions decreased across sessions for decoding models exhibiting improved performance, such as those distinguishing between social cues (Fig. 4f,g and Extended Data Fig. 9a,b). By contrast, a decrease in weight variance, kurtosis or skewedness was not observed in models in which decoding performance did not improve, such as when distinguishing between non-social cues (Fig. 4g) or between categories and choice in V4. These findings were consistent across brain areas and were present in each monkey (Extended Data Fig. 9b). The decrease in these weight metrics indicates that early during learning, a select number of cells contribute most to social interactions relative to the rest of the population. However, although learning progressed, the magnitude of the weights decreased, and information about social events became distributed more evenly across the population (Fig. 4f and Extended Data Fig. 9a).

## Learning improves spiking coordination

Temporal coordination of neuronal spiking is believed to be correlated with neuronal communication and information flow[38,39]. We computed spike time correlations between pairs of cells in V4 and dlPFC and across areas. Only significant CCGs were used in the analysis (the peak of shuffle-corrected CCGs was greater than 4.5 standard deviations of the tails[40]; Methods). To account for the delay in information transmission within and between areas, we only analysed within-area CCGs (average of 668 cell pairs per session) that peaked at ±0–6 ms time lag and interareal CCGs (average of 45 pairs per session) that peaked at ±15–60 ms lag (Fig. 5a). The mean coordination, or the average of significant cell pair maximum coincident spikes (CCG peaks), significantly increased across sessions during social events (Fig. 5b,c). Improved spike-timing coordination was not due to individual neurons' time locking with respect to social events during learning, reflecting communication within and across cortical areas (Extended Data Fig. 10b). In V4, pairwise synchrony increased by 114% across sessions during fixations on social cues but not before choice events, whereas in dlPFC, synchronized spiking increased on average by 137% during fixations on social cues

and self-choice but not during partner-choice. As animals learned to cooperate, coordination between V4 and dlPFC increased by 160% for all social events except partner-choice (Fig. 5b,c). Importantly, both within and between areas, coordination during fixations on random floor objects or random time periods (for example, intertrial interval) did not change across sessions, indicating that increased coordination exclusively correlates with learning of social events (Fig. 5b and Extended Data Fig. 10a). Additionally, a significant number of V4–dlPFC pairs exhibited maximum coincident spiking at both positive (V4 leads) and negative (dlPFC leads) time lags during social events, indicating that V4–dlPFC coordination reflects both feedforward and feedback interactions.

Finally, we asked whether the V4 and dlPFC cells that were coordinated in their spike timing would also exhibit improved encoding of social variables. Notably, significantly correlated pairs of V4–dlPFC neurons contributed more to the encoding of social events in each brain area, as their normalized weight values from decoding models for social cues and choice were significantly higher than the weights of the remaining population (Fig. 5e; Wilcoxon signed-rank test, $P < 0.01$). In V4, this result applied for the encoding of social cues but not choice, whereas in dlPFC it applied to all events except partner-choice (Fig. 5e). Taken together, we propose a general mechanism for learning social interactions whereby increased spike timing coordination between areas V4 and dlPFC during social events leads to improved encoding and distributed representation of social variables within each area (Fig. 5f).

## Discussion

Social interactions, especially cooperation, require an interpretation and exchange of sensory information, including relevant visual cues, among engaging agents. However, technological limitations have prevented an understanding of how visual information is encoded and passed on to executive areas to guide social decisions in freely moving animals. Our results show that across sessions, animals are more likely to engage in cooperation after viewing social cues. This is supported by increased coordinated spiking between visual and prefrontal cortical neurons during learning to cooperate, which is associated with improved accuracy of neural populations to encode social cues and the decision to cooperate. This provides the first evidence, to our knowledge, of the role of the visual cortex in encoding socially relevant information. Somewhat surprisingly, dlPFC neurons outperformed those in V4 in their ability to discriminate between several visual social cues, probably because dlPFC receives and integrates diverse sensory modalities[25,41], which may enable a better prediction of highly dimensional incoming stimuli. We further discovered that early during learning, a select number of cells contributed most to social interactions compared to the rest of the population. However, although learning progresses, the information about social and decision-making signals became more evenly distributed across the neural population.

We propose that learning cooperation emerges from improved population coding and communication between visuo-frontal circuits during social viewing. Notably, the strongest coupled neurons across areas were those that contributed the most to the encoding of social events. As animals acquired cooperation behaviour, they increased the viewing of social cues before deciding to cooperate (Figs. 2d and 3c). This raises the possibility that during task-relevant events, increased spike timing coordination between visual and prefrontal cortical neurons reflects strengthened synapses between cells[42,43] or aligned communication subspaces (that is, firing rate patterns) between neuronal populations in each area[44,45]. Surprisingly, an increase in spiking coordination was not observed in dlPFC or between V4–dlPFC before the conspecific's choice (Fig. 5b,c), which indicates that dlPFC may encode the prediction of the other's behaviour in mean firing rates but not spike timing

coordination. We further suggest that increased spiking coordination may also occur between dlPFC and other brain areas during allocentric events, which could be explored in future studies[7,19,46].

Finally, by allowing animals to move freely during social cooperation, our study represents a move towards studying the neural underpinnings of naturalistic behaviour in a free-roaming setting. Although this paradigm shift has long been suggested[47,48], recent advances in low-power, high-throughput electrophysiological devices coupled with wireless behavioural monitoring and large-scale computing[36,49,50] made this research feasible only now. Critical to our work is our simultaneous use of wireless neural and eye-tracking recordings to examine how visual events and social cues guide the decision to cooperate. Analysing the relationship between the behavioural repertoires of each freely interacting agent allowed us to uncover the neural computations involving prioritization of social visual cues that were essential to social learning. Thus, vision may be the social language of primates, probably governing learning of various social activities such as grooming, play and collective foraging. Future research will investigate how other sensory cues, such as odours, vocalizations and touch[51,52], complement visual information to guide neural processes underlying social decisions. A shift towards more natural behaviour in which multisensory information is recorded wirelessly in conjunction with large-scale population recordings will be essential for understanding the neural mechanisms of social cognition[53,54].

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

## Methods

### Animals

All experiments were performed under protocols approved by the University of Texas at Houston Animal Care and Use Committee and the Institutional Animal Care and Use Committee for the University of Texas Health Science Center at Houston (UTHealth). Four adult male rhesus monkeys (*Macaca mulatta*; selfM1: 10 kg, 11 years old; partM1: 12 kg, 10 years old; selfM2: 14 kg, 12 years old; partM2: 12 kg, 16 years old) were used in the experiments.

### Behavioural training and experiments

Before starting learning cooperation experiments, we trained individual monkeys in their home cage to press a button resembling that in the cooperation arena to receive a food reward. Therefore, when beginning social experiments, each monkey was familiar with the button and button press–reward association. The cooperation arena measured 7 × 4 × 3 ft (length × width × height) and was constructed out of polyvinyl chloride piping, plastic and plexiglass. Animals were acclimated to the arena and button before beginning social learning experiments. Buttons in the arena were strategically placed next to the clear divider so that monkeys could easily see each other's actions. During cooperation experiments, animals had to learn to push and hold buttons simultaneously with their partner-monkey to receive reward. Each animal's tray contained their own amount of food reward (banana-flavoured pellet; not accessible by the other monkey), and the trays moved together at the same speed while animals were simultaneously pressing. If one monkey stops pressing, the trays stop at their location on the track and continue moving forward only when both monkeys begin pressing again. We used the term 'intertrial interval' for the 20 s period consisting of a 10 s pause after a trial ends followed by tray movement back to the pellet dispensers (approximately 5 s) and another 5 s pause before the next trial begins. Trial durations ranged from 10 s to 30 min, depending on how long it took for animals to cooperate. Although learning sessions consisted of 100–130 trials, the behavioural events examined ('partner-push', 'self-push', 'fixations on reward', 'fixations on partner') often occurred more than once per trial. For complete quantification of events, see 'Neural responses during social cooperation'. For non-social control experiments, after learning cooperation sessions, we recorded control sessions in which animals completed the cooperation task with an opaque divider separating them, so animals could no longer see each other but could still hear and smell one another. The opaque divider was a thin (6 mm thickness), opaque grey piece of plexiglass and was placed over the clear plexiglass that already divided the animals. The end of the plexiglass nearest the buttons protruded out of the arena to ensure that animals could not see each other pressing. We recorded four control sessions with the opaque divider for each monkey pair. Opaque-divider sessions had the same number of trials with the same trial structure as regular learning cooperation experiments. The first opaque-divider session completely confused the animals as they did not even attempt to push after reward dispensed. However, they did eventually complete trials together by using the 'click' sound of the button when it is pressed. Sometimes one monkey would push the button many times to signal their partner to push, but cooperation performance in these sessions was significantly impaired (Extended Data Fig. 6a). Although macaques do vocalize, they did not produce vocalizations during these experiments and did not seem to use vocalizations to coordinate behaviour.

Solo-and-social control experiments (Extended Data Fig. 6b–d) occurred after learning experiments and included periods of solo trials in which the self-monkey was alone in the arena, pellets dispensed only in their tray, and the monkey pushed their button to deliver reward. We recorded nine more sessions using these 'solo blocks' whereby the self-monkey completed sets of cooperation trials entirely by themselves. Solo blocks were interspersed with regular cooperation blocks conducted with two animals. Solo-and-social sessions included 60 solo and 60 social trials that were executed in an alternating block design to eliminate confounds of recording quality or stability in a session. We alternated 30 solo trials, 60 social trials, 30 solo trials and vice versa between days.

### Chronic implantation of the Utah array

A titanium headpost was implanted medially with anchor screws. After acclimatization and behavioural training (described in the previous section), animals were implanted with a 64-channel dual Utah array in the left hemisphere dlPFC (anterior of the arcuate sulcus and dorsal of the principal sulcus) and left V4 (ventrally anterior to lunate sulcus and posterior to superior temporal sulcus) and a pedestal on the caudal skull (Blackrock Neurotech). We used Brainsight, a neuronavigational system, and the animals' magnetic resonance imaging scans to determine the location for V4 and dlPFC craniotomies (Rogue Research). During surgery, visual identification of arcuate and principal sulci guided precise implantation of arrays into the dlPFC, and visual identification of the lunate and superior temporal sulci supported array placement in area V4. The dura was sutured over each array, and two reference wires were placed above the dura mater and under the bone flap. Bone flaps from craniotomies were secured over the arrays using titanium bridges and screws. After the implant, the electrical contacts on the pedestal were always protected using a plastic cap except during the experiment. Following array implantation, animals had a three-week recovery period before recording from the arrays.

### Wireless electrophysiology

To record the activity of neurons while minimizing interference with the animals' behaviour, we used a lightweight, rechargeable battery-powered device (Cereplex-W, Blackrock Neurotech) that communicates wirelessly with a central amplifier and digital processor (Cerebus Neural Signal Processor, Blackrock Neurotech). First, the monkey was head-fixed, the protective cap of the array's pedestal was removed and the wireless transmitter was screwed to the pedestal. Neural activity was recorded in the head-fixed position for 10 min to ensure the quality of the signal before releasing the monkey in the experimental arena. The arena was surrounded by eight antennas. Spikes from each brain area were recorded simultaneously at 30 kHz and detected online (Cerebus Neural Signal Processor, Blackrock Neurotech) using a manually selected upper and lower threshold on the amplitude of the recorded signal in each channel, which was helpful to eliminate noise from the animal chewing, or by using the software's automatic thresholding, which was ±6.25 times the standard deviation of the raw signal. The onsite digitization in the wireless device showed lower noise than common wired head stages. The remaining noise from the animal's movements and muscle activity was removed offline using the automatic algorithms in Offline Sorter v.4 (Plexon). In brief, this was done by removing the outliers (outlier threshold = 4–5 standard deviations) in a three-dimensional (3D) space that was formed by the first three principal components of the spike waveforms. Then, the principal components were used to sort single units using the *k*-means clustering algorithm. Each signal was then automatically evaluated and manually checked as multi- or single-unit using several criteria: consistent spike waveforms, waveform shape (slope, amplitude, trough-to-peak) and exponentially decaying interspike-interval histogram with no interspike interval shorter than the refractory period (1 ms). The analyses here used all single- and multi-unit activity.

### Receptive field mapping

We identified receptive fields of recorded neurons in a head-fixed task in which the animal was trained to maintain fixation during stimulus presentation on a monitor. Neural activity was recorded and thresholded using a wired head-stage and recording system, similar to the methods described above in 'Wireless electrophysiology' (Cerebus Neural Signal Processor, Blackrock Neurotech). We divided the right visual field into a

3 × 3 grid consisting of nine squares with each square covering 8 × 8° of visual space. The entire grid covered 24 × 24° of visual space. Each of the nine squares was further subdivided into a 6 × 6 grid. In each trial, one of the nine squares was randomly chosen, and the receptive field mapping stimuli were presented at each of the 36 locations in a random order. The receptive field mapping stimuli consisted of a reverse correlation movie with red, blue, green and white patches (approximately 1.33° each). A complete receptive field session is composed of ten presentations of the receptive field mapping stimuli in each of the nine squares forming the 3 × 3 grid. We averaged the responses over several presentations to generate receptive field heatmaps and corresponding receptive field plots (as shown in Fig. 3b). As recorded populations remained stable across days (Extended Data Fig. 3), receptive field mapping was done before starting learning sessions and performed once every month during recordings.

## Wireless eye tracking

We used a custom wireless eye tracker (ISCAN) to measure pupil position and diameter from the self-monkey during experiments. The portable wireless eye tracker, mounted dorsally, right above the animal's head, consisted of an eye mirror, an eye camera for detecting pupil size and eye position, and a scene camera situated above the eye camera (see also ref. 50), which records animal's field of view. All data were recorded at 30 Hz. To train animals to wear the device without damaging it, its 3D geometry was modelled (Sketchup Pro), and dummies were 3D-printed and fitted with eye mirrors. To properly position the eye tracker and dummies relative to the eye, custom adaptors were designed and 3D-printed to attach directly to the animal's headpost and serve as an anchor point for the eye tracker. These adaptors were designed to interface with the headpost without touching the animal directly, to minimize discomfort and reduce the likelihood of the device being tampered with. These dummy eye trackers were worn by animals for several mock recording sessions to adjust them to wearing the device. Once the animals grew accustomed to wearing the dummy and stopped touching it altogether, the real device was used. Before each experiment, the eye tracker was secured on the animal and we performed a calibration procedure ('point-of-regard' calibration) while the animal was head-fixed, which mapped the eye position data to the matrix of the head-mounted scene camera (Extended Data Fig. 1a). Animals were trained to view five calibration points within the field of view of the scene camera (640 × 480 pixel space of a scene camera frame maps to 35 × 28°, length × height), including a centre calibration point and four outer points positioned at ±8 to ±10° with respect to the centre (we chose the distance between the animal's eye and calibration monitor on the basis of the approximate range of eye-stimulus distances during free viewing, which was 70 cm). As the animal viewed each point, the eye-calibration software synchronized the eye-movement data with the image frames recorded by the scene camera. After calibration, the animal's centre of gaze is displayed on each scene camera frame in real time as a crosshair (Figs. 1d and 2a). If the animal looks outside the field of view of the scene camera frames, gaze location is not detected and the crosshair will not appear on the scene camera frames. When this occurs, eye position data are reflected as zero (Extended Data Fig. 1c). Therefore, only scene camera frames that included a crosshair were used in analysis, which usually occurred (60–85%; Extended Data Fig. 1e). We used the horizontal and vertical coordinates of the pupil to compute eye speed. To extract fixations, we used movement-threshold identification to determine the conservative speed threshold that best separated the bimodal distribution of eye speed in a session[55]. A fixation was defined as a minimum 100 ms period when the eye speed remained below this threshold.

## Behavioural tracking

We captured a top-down, or overhead, video of the animals during the experiments using a COSOOS CCTV LED home surveillance security camera. We recorded overhead and scene camera (wireless eye tracker) videos using a CORENTSC DVR (I.O. Industries). This DVR recorded all videos at 30 frames per second and sent pulses to the Blackrock Cerebus neural recording system for every captured frame from each camera, as well as the start and end of video recording. We used the timestamps of these pulses to synchronize overhead video frames to neural and behavioural data. Owing to imperfect transmission of wireless eye-tracking data, frames were sometimes dropped from the recording. Therefore, to verify the timestamp of each scene camera frame, we used a custom object character recognition software (developed by S. Yellapantula) to automatically read the timestamp listed on each scene camera frame and align with neural data. Using DeepLabCut v.2.0, we trained a network to automatically label relevant objects in the frames, such as the crosshair, reward dispensers and trays, each animal's button and various body parts of the partner-monkey including eyes, head, ears, nose, shoulders, limbs, chest, back, face, paws and butt (Extended Data Fig. 1d). The DeepLabCut output included the location coordinates of all the objects found in the frames. Therefore, we used a degree-to-pixel conversion and the coordinates of the crosshair and object labels to identify which objects were in the receptive fields of the neurons in any given frame (Extended Data Fig. 1b; 5° = 90 × 80 pixels, on the basis of the equation $\text{Tan}\,\varnothing = d \div x$, where $d$ is the measured length and height of the scene camera frame when viewing from distance $x$).

Finally, to compute head, limb and torso speed (movement) of the animal, as shown in Extended Data Fig. 5, we used DeepLabCut to label body parts of the self-monkey in the overhead camera frames. 'Head movements' included labels from the centre of the head, snout and each ear. 'Limb or arm' movement was computed from shoulder, elbow and paw labels. 'Torso or whole-body' movement was calculated from the animal's upper and mid back labels. To compute the average speed of each label during frames of interest, we calculated the Euclidean distance between the label coordinates across consecutive frames. Subsequently, we quantified the overall movement of each body area (head, limb or torso) by averaging the speeds of the corresponding labels.

## Conditional probability

For each trial in a session, we computed the conditional probability of cooperating for the self- and partner-monkey, respectively, using the equations

$$P(\text{Self}|\text{Part}) = \frac{P(\text{Self and Part})}{P(\text{Part})} \text{ and } P(\text{Part}|\text{Self}) = \frac{P(\text{Self and Part})}{P(\text{Self})}$$

The probabilities were derived from button-push sequences for each monkey that were represented as a time series of zeros and ones in 100 ms bins. Conditional probabilities were averaged across trials in a session (100 in monkey pair 1 and 120 in monkey pair 2) to create the values plotted in Fig. 1h.

## Markov model

To explore the relationship between fixation and push events, we used a Markov model to estimate the transitional probabilities between social events as they occurred in a sequence across a trial, using the function hmmestimate in MATLAB 2020b. Sequences consisted of four events/states: 'view reward', 'view partner', 'self-push' and 'partner-push', resulting in 16 event pairs and transitional probabilities. We only included trials in which all four events occurred, which was on average 40% of trials per session. For each event pair, transitional probabilities were averaged across trials for a session mean transitional probability, as seen in Fig. 2d. Transitional probabilities for each session and each monkey pair are shown in Extended Data Fig. 2.

## Identifying stable units across sessions

We used principal component analysis (PCA) of waveforms from each session to identify stable spike waveforms across sessions. First, we

performed PCA on a matrix of 100 samples of waveforms of single units and multi-units from every session using the pca function in MATLAB 2020b. Then, for each session, we used the first seven components of the principal component coefficients to compute the Mahalanobis distance between distributions of all waveforms for each combination of cell pairs in that session. The Mahalanobis distance between two objects in a multidimensional vector space quantifies the similarity between their features as measured by their vector components. It takes into account the covariance along each dimension[56]. The Mahalanobis distance between two clusters of spike waveforms $A$ and $B$ belonging to a pair of neurons in seven-dimensional vector space was computed using the following formula:

$$MD = \sqrt{((A-B)^T V^{-1} (A-B))} ,$$

where $T$ represents the transpose operation, $V^{-1}$ is the inverse of the covariance matrix and $A$ and $B$ are the first seven components of the principal component coefficients for each neuron in the pair. Importantly, because this analysis was performed among cell pairs in a session, this distribution reflects the Mahalanobis distances for distinct, individual cells. We combined the distances across sessions to create one distribution and used this to identify a waveform threshold, which was the fifth percentile of the distribution. Therefore, Mahalanobis distances between waveforms of cell pairs that are less than the threshold reflect waveform distributions that belong to the same neuron.

For each channel (electrode), we computed the Mahalanobis distance using PCA waveform coefficients from all the neurons identified on that channel across all sessions. Some channels recorded two units each day, and others did not record any isolated single-unit activity or multi-unit activity. Ninety-six electrodes were recorded from each subject's brain; however, electrodes without single units or multi-units for at least ten (that is, half) of the learning sessions were not used in the analysis. Analysis included 90 total electrodes from M1 and 86 total electrodes from M2. Mahalanobis distance values that were less than the threshold represented stable units, and cell pairs whose Mahalanobis distance was above the threshold indicated that the same neuron was not recorded across both sessions. Channels with stable units and a channel with stable multi-unit activity but unstable single-unit activity are shown in Extended Data Fig. 3a. The number of stable cells divided by the total number of cells is the percentage of stable units in each area for each monkey, as shown in Extended Data Fig. 3b. In monkey 1, 81% of recorded units (504/620) in V4 and 74% of recorded units in dlPFC (1,350/1837) were consistent across sessions. In monkey 2, 83% of recorded units in V4 (1,479/1,773) and 71% of recorded units in dlPFC (561/794) were stable (Extended Data Fig. 3b). Overall, our analysis yields results comparable to other electrophysiological studies with Utah array recordings, which also found that chronically implanted Utah arrays typically record from the same neurons across days and months[29,49,57]. The neural analyses in Figs. 4 and 5 were repeated using only the units that remained stable across sessions (Extended Data Fig. 7), and the main results remain unchanged from those using the entire population (Figs. 4 and 5).

### Neural responses during social cooperation

We identified four salient events for cooperation: 'fixations on the reward', 'fixations on the partner-monkey', 'self-pushes' and 'partner-pushes'. 'View reward', or 'fixations on the reward', includes fixations on the food reward system: the pellet dispenser and tray. (Note that the tray always contained pellets during tray fixations. The only time a tray did not contain food reward was during the intertrial interval, and we did not include fixations from this period in the analysis.) 'View partner', or 'fixations on the partner', includes all fixations on the partner-monkey (head and body). 'Social cues' includes all fixations categorized as 'view reward' and 'view partner'. On average, there were 826 fixations on the reward, 936 fixations on the partner, 116 self-pushes

and 43 partner-pushes (lead trials only; Extended Data Fig. 4a) per session for each animal pair. To determine whether a cell was significantly responding to one or more of these events, we compared the firing rate in a baseline period (intertrial time, specifically 4.5 seconds before trial start) to the event onset using a Wilcoxon signed-rank test followed by FDR correction. Specifically, for each neuron, we calculated its firing rate (20 ms bins) occurring 130 ms after fixation onset that accounted for visual delay (60 ms for V4 neurons and 80 ms for dlPFC). We chose this window as the fixation response period because most of our fixations were 100–200 ms in duration (Extended Data Fig. 1f). For self- and partner-push, we used 100 ms bins to compute the firing rate and 1,000 ms before push onset as the response period because firing rates began to significantly increase during this time. For partner-pushes on 'partner lead' trials only (Extended Data Fig. 4a), we used 500 ms before and 500 ms after the push, because the self-monkey viewed them after this push (trays were not moving as the self-monkey was not yet pushing in these types of observations). Neural activity occurring between the moment trays began moving and the end of a trial was never used in any analyses in this Article. Additionally, non-fixation rewarding events, such as the start of the trial when pellets were dispensed and the end of the trial when pellets were received, were not included in any analyses. For each neuron, response firing rates were compared to baseline firing rates that were computed across the same duration as social event responses (130 ms for fixations and 1,000 ms for pushes). Some recorded cells did not respond significantly to any social events, known as 'other' (Fig. 3e, right). The percentage of neurons responding to social events did not systematically differ across sessions (Extended Data Fig. 3c).

### d prime

To assess the discriminability of neural responses between different stimuli, we computed $d$ prime ($d'$) following established methods in neuroscience[58]. $d$ prime is a widely used measure that quantifies the signal-to-noise ratio in a discrimination task, indicating the ability of a neuron to distinguish between two classes of stimuli. In a session, for each neuron, the mean responses ($u_A$ and $u_B$) to stimuli A and B as well as the standard deviations of responses were obtained. We computed $d'$ using the following formula:

$$d' = \frac{u_A - u_B}{0.5 \times \sqrt{\sigma_A^2 + \sigma_B^2}}$$

Here, $u_A$ and $u_B$ are mean responses (averaged across response time for each event, described above) of one neuron for each trial to stimuli A and B, respectively. Stimuli A and B were those used in decoding models ('self-push' or 'partner-push', 'view reward' or 'view partner', 'view button' or 'view random' and social and non-social categories; see Extended Data Fig. 9c,d).

### SVM decoder

We used a SVM decoder[59] with a linear kernel to determine whether the population firing rates in V4 or dlPFC carry information about visual stimuli and/or decision-making (Fig. 4). Specifically, we computed the mean firing rates of each neuron in the population for the response period (described above) in each observation of fixations or pushes (observations of any social event could occur in one trial) and then classified binary labels specifying the event (for example, fixations on reward were class one, fixations on partner were class two) from neural responses. For each session, the number of fixations or pushes was always balanced across classes. Random selections of class observations were repeated for 100 iterations, giving us the average classification accuracy over 1,000 test splits of the data for each session. To train and test the model, we used a tenfold cross-validation. In brief, the data were split into ten subsets, and in each iteration the training consisted of a different 90% subset of the data; the testing was done

with the remaining 10% of the data. We used the default hyperparameters as defined in fitcsvm, MATLAB 2020b. Decoder performance was calculated as the percentage of correctly classified test trials. We compared model performance for predicting train and test data to check for overfitting. In each session and iteration, we trained a separate decoder with randomly shuffled class labels. The performance of the shuffled decoder was used as a null hypothesis for the statistical test of decoder performance.

For improved data visualization in Fig. 4, we plotted the shuffle-corrected decoder accuracy (actual − shuffled decoder performance), but learning trends remain even when only the actual decoder accuracy is evaluated (Extended Data Fig. 8a). Animals cooperate more quickly as they learn, and sessions become shorter, so the number of observations typically decreases across sessions. However, this change in the number of observations across sessions did not influence decoding performance. We repeated the analysis in Fig. 4a–d, for which we balanced the number of fixations between classes and across sessions. For each brain area, decoding accuracy was comparable to the original and still significantly improved during learning (Extended Data Fig. 8b).

For the Fig. 4a–c analyses, 14 sessions were analysed from pair 2 because of an inadequate number of fixations on the stimuli in three of the 17 sessions. Similarly, for Fig. 4b, only 16 sessions were included in the analysis because monkey 1 did not fixate enough on the self-button during two sessions. Sessions with fewer than 30 fixations were not included in any neural analyses. For the analysis in Fig. 4e, the number of observations matched for 'with cue' and 'without cue' classes to enable fair comparison of decoder performance across conditions. Note that in Fig. 4e, V4 and dlPFC accuracies (navy and gold) are different than those in Fig. 4d because this analysis always included pushes with preceding fixations, whereas the Fig. 4d analysis used pushes with or without preceding fixations.

For comparing feature weights of correlated and non-correlated V4 and dlPFC neurons (Figs. 4f,g and 5e and Extended Data Fig. 9), we first normalized weights across the entire population of neurons in each session[37] using the equation below, where $W_o$ is the current cell weight divided by the square root of the sum of all the squared weights in the population. $n$ is the cell number:

$$\text{normalized weight} = \frac{W_o}{\sqrt{\sum_{i=1}^{n} W_n^2}}$$

## Cross-correlation

CCGs of the animals' actions in Fig. 1 were computed using the animals' button-push sequences occurring across a trial, represented as a series of zeros and ones in 100 ms time bins. For each cooperation trial in a session, push series for each monkey (sequences were of equal length) were cross-correlated using the xcorr function in MATLAB 2020b. Coefficient normalization was used, which normalizes the sequence so that the autocorrelations at zero lag equal 1. The cross-correlations were averaged across trials to create a session CCG, as plotted in Fig. 1f. The maximum value, or peak, of each session's CCG is plotted as the mean coordination for that session, as shown in Fig. 1g. The time lag at which the peak occurred in each session is the push lag, shown in Fig. 1f, right. Another 'shuffled' analysis was performed for comparison, in which the push sequences derived for each monkey were shuffled randomly in time for each trial. Trial cross-correlations between animals' shuffled pushes were calculated and then averaged across trials to create a session CCG of shuffled presses, as shown in Fig. 1f. As with the actual CCGs, the peak of each session's shuffled CCG is plotted as the mean coordination for that session and shown in Fig. 1g.

CCGs in Fig. 5 were computed by sliding the spike trains of each cell pair and counting coincident spikes in 1 ms time bins for each social event and pair of neurons (within and between areas) using the xcorr function in MATLAB 2020b. Cross-correlations were normalized by the geometric mean spike rate to account for changes in individual neurons' firing rates and further corrected for stimulus-induced correlations by subtracting an all-way shuffle predictor, efficiently computed from the cross-correlation of the peristimulus time histograms[38,40,60]. Specifically, the trial-averaged cross-correlation of the binary time series spike trains between neurons $j$ and $k$ was computed as

$$C_{jk}(\tau) = \frac{1}{M} \sum_{i=1}^{M} \sum_{t=1}^{T} x_j^i(t) x_k^i(t+\tau), \tag{1}$$

where $M$ is the number of trials, $T$ is the duration of the spike train segments, $x$ is the neural response and $\tau$ is the lag. We normalized the above trial-averaged cross-correlation (equation (1)) by dividing it by the triangle function $\Theta(\tau)$ and the geometric mean of the average firing rates of the neurons $\sqrt{\lambda_j \lambda_k}$ (ref. 40) to get the unbiased CCG of the spike trains in units of coincidences per spike:

$$\text{CCG}(\tau) = \frac{C_{jk}(\tau)}{\Theta(\tau) \sqrt{\lambda_j \lambda_k}}. \tag{2}$$

The function $\Theta(\tau)$ is a triangle representing the extent of overlap of the spike trains as a function of the discrete time lag $\tau$:

$$\Theta(\tau) = T - |\tau| \quad (-T < \tau < T), \tag{3}$$

where $T$ is the duration of the spike train segments used to compute $C_{jk}$ (ref. 40). Dividing $C_{jk}$ by $\Theta(\tau)$ corrects for the triangular shape of $C_{jk}$ caused by the finite duration of the data[40]. Dividing by $\sqrt{\lambda_j \lambda_k}$ in equation (2) results in CCG peaks with relatively constant area as firing rates of individual neurons change[40,61]. In other words, dividing by the geometric mean of the firing rates of the two neurons makes the CCG peaks relatively independent of the firing rates.

We computed CCGs using spiking activity that occurred 800 ms before choice or random events and 200 ms after fixation onset with visual delay ('Neural responses during social cooperation'). For cross-correlation of V4−dlPFC responses to fixations, we used an 80 ms visual delay. A CCG was considered significant if the peak (occurring within a −6 to +6 ms lag interval within area and ±15−60 ms lag interval between areas) exceeded 4.5 times the standard deviations of the noise (tail) level occurring ±60 ms from the peak range during non-fixation events and ±25 ms from the peak range for fixation events. Mean coordination values for each session are the average of the CCG peaks of all significant cell pairs. For random events, we used times from the inter-trial period, and for random fixations, we used fixations on objects that were not social cues. In a session, the number of random observations matched those of social events. Mean coordination values for monkey pair 2 are in Extended Data Fig. 10a.

## Statistics

To assess systematic changes in behavioural and neural metric performance or learning, we report the $P$ value from simple linear regression and Pearson's correlation coefficient to report the strength and direction of linear relationships. The per cent increase or decrease of behavioural and neural metrics was calculated by the percent change equation, $C = \frac{x_2 - x_1}{x_1}$, where $C$ is the relative change, $x_1$ is the value from session 1 and $x_2$ is the value from the last session. Changes were then averaged across events or monkeys. For comparing two paired groups such as a cell's firing rate during an event and a baseline period, we used the two-sided Wilcoxon signed-rank test. We chose this test rather than parametric tests, such as the $t$-test, for its greater statistical power (lower type I and type II errors) when data are not normally distributed. When multiple groups of data were tested, we used the FDR multiple-comparisons correction, whose implementation is a standard function in MATLAB. When comparing two unpaired distributions, we used the Wilcoxon rank-sum test.

**Reporting summary**

Further information on research design is available in the Nature Portfolio Reporting Summary linked to this article.

## Data availability

All source data used to generate experimental figures are available at https://zenodo.org/records/10384447. The data that support the findings of this study are available from the corresponding authors upon reasonable request.

## Code availability

Data analysis was performed using MATLAB 2020b (MathWorks). The code on which this study was based is available from the corresponding author upon reasonable request.

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

**Acknowledgements** This work was supported by the National Institutes of Health BRAIN Initiative grant nos. U01NS108680 (B.A., A.W. and V.D.) and 1F31MH125451 (M.F.). We thank N. Shahidi and A. Jones for their electrical design and software programming of the cooperation paradigm and S. Pojoga for programming work. Figure 1a,b and Extended Data Fig. 1a (monkey head) were created with Biorender.com, using images provided by Blackrock Neurotech and with design ideas from A. McConnell. The neural transmitter and Utah array images in Fig. 1b were provided by Blackrock Neurotech. The wireless eye tracker image in Fig. 1c was designed by A. Parajuli. The brain in Fig. 5f was drawn by A. Andrei.

**Author contributions** A.W., V.D. and B.A. conceptualized the research. M.F. designed the methodology and software. M.F., S.Y., A.P. and N.K. analysed the data. M.F. V.D., B.A., A.W. and M.F. provided resources for the investigation. M.F., A.P., S.Y. and N.K. curated the data. M.F. and V.D. wrote the original draft of the paper and reviewed and edited it. M.F., S.Y., A.P., N.K. V.D., B.A. and A.W. supervised the project, with project administration by V.D. and M.F. Funding acquisition was performed by V.D., B.A., A.W. and M.F.

**Competing interests** The authors declare no competing interests.

**Additional information**
**Correspondence and requests for materials** should be addressed to Valentin Dragoi.

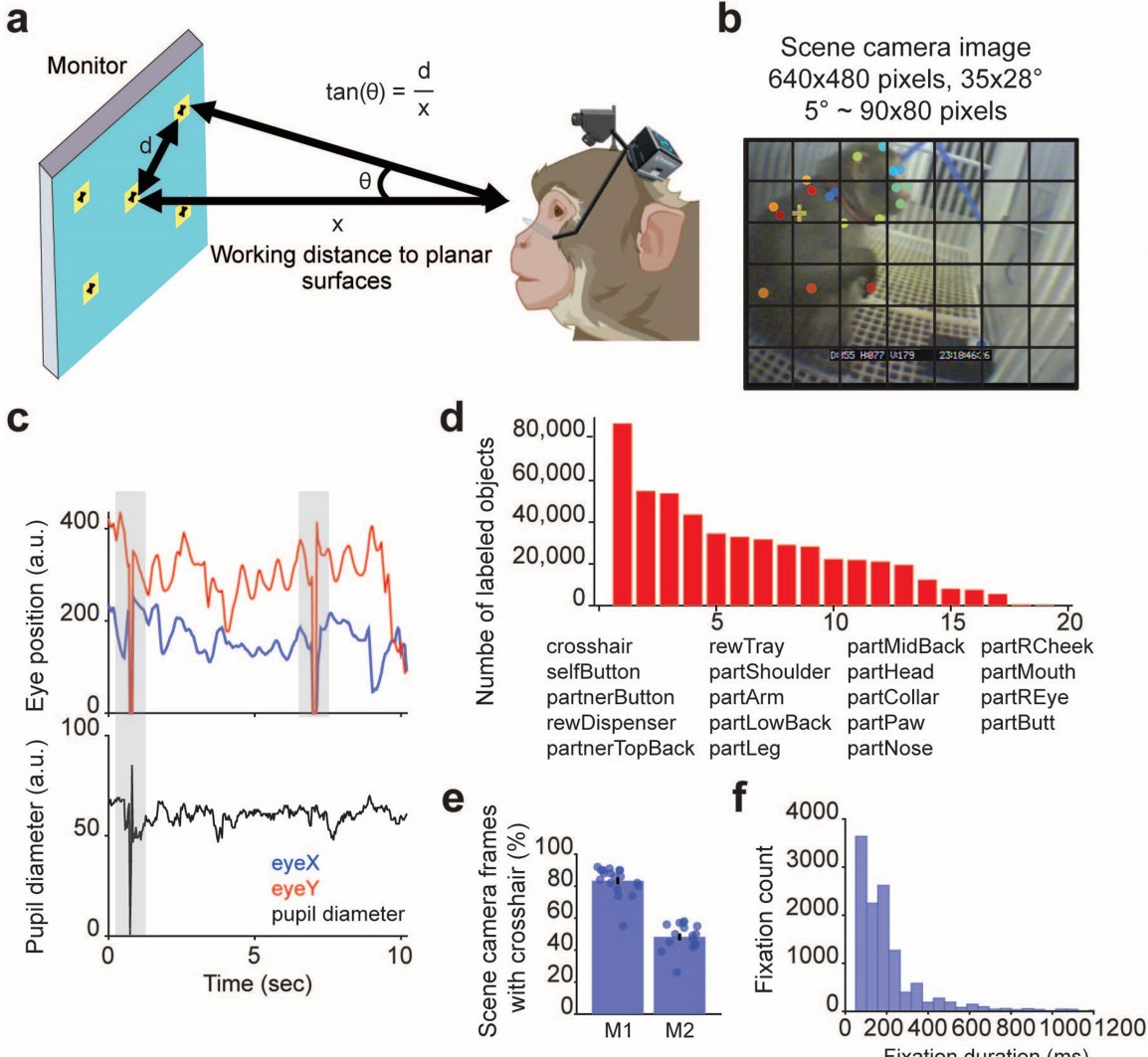

**Extended Data Fig. 1 | Wireless eye tracking methods and fixation statistics. a**, Eye tracking calibration procedure. As the animal views five points on a monitor, this information is entered into the program (ISCAN Inc.), which projects a crosshair indicating the animal's point of gaze onto scene camera frames. **b**, Using the equation in panel a, pixel space of the scene camera is converted to degrees to identify when objects in the scene camera frames are within the receptive fields of neurons. Here, the animal's shoulder and upper arm are within receptive fields. **c**, Raw traces of eye x and y coordinates, and pupil diameter recorded with the wireless eye tracker. The zero values at 1 second are due to a blink, while the zero values of x and y coordinates at

7 seconds are due to the animal viewing an object located out of the field of view captured by the scene camera. **d**, Number of objects (sorted) that DeepLabCut labeled in the scene camera frames from one session. **e**, Session-averaged percentage of scene camera frames out of total recorded that contained the crosshair for each monkey. M1: 2382652 frames labeled out of 2844338 total frames. M2: 1158612 frames labeled out of 2421325 total frames. Each circle is the percentage of crosshair labeled frames for each session. **f**, Histogram of fixation durations from one representative session that consisted of 12,378 fixations. 70% of the fixations were 200 ms duration or less. Illustrations in **a** were created using BioRender.

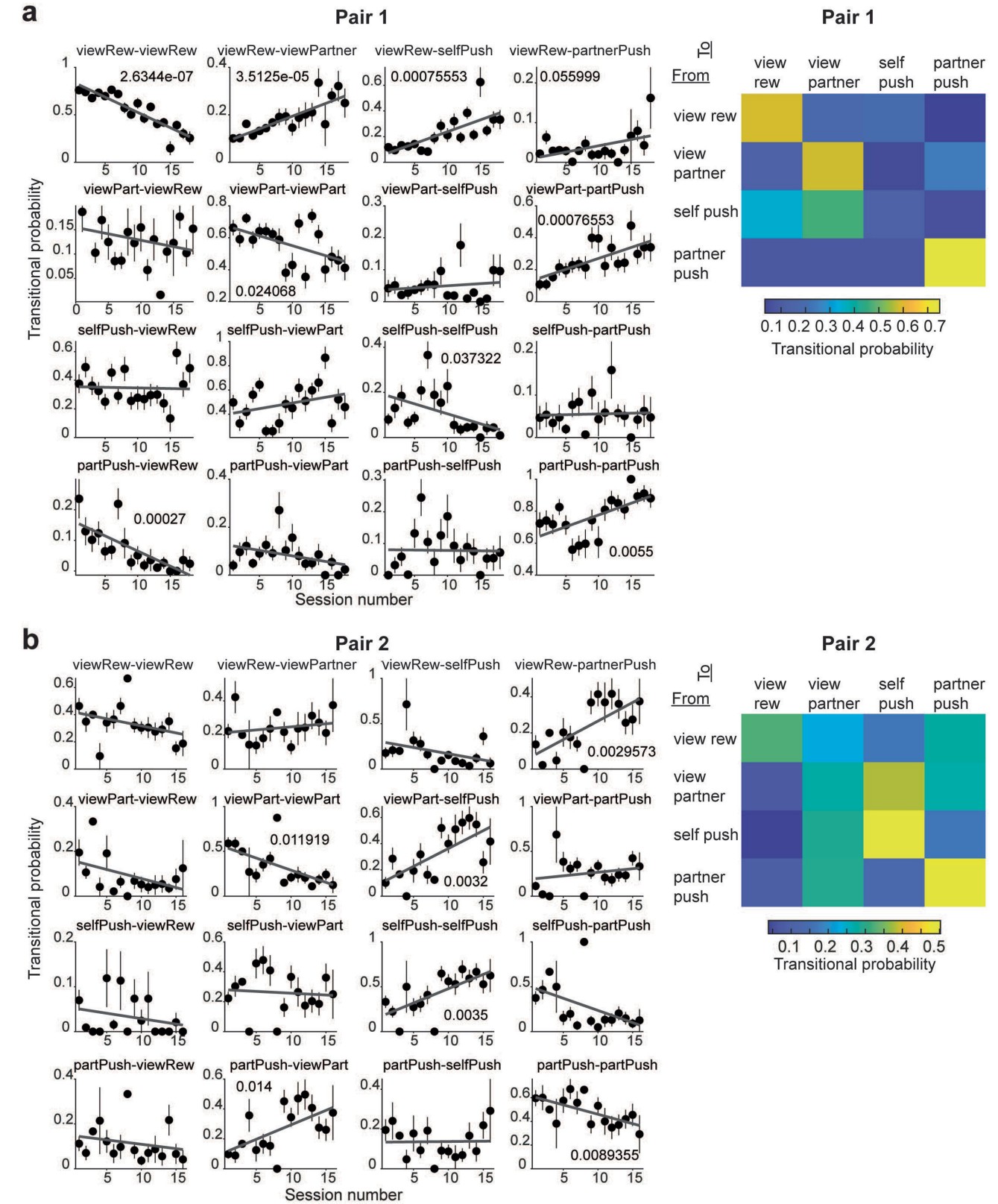

**Extended Data Fig. 2 | Markov Model transitional probabilities between social events for each monkey pair. a**, Left - Transitional probabilities from Markov Modeling estimation, plotted across sessions for each event pair combination in monkey pair 1. The P value is included if simple linear regression P < 0.05. Across monkeys, most increasing trends occur for event pairs that begin with or include a viewing behavior. Right – the transitional probability matrix for all event pairs, averaged across sessions. **b**, Same as in a, but for monkey pair 2.

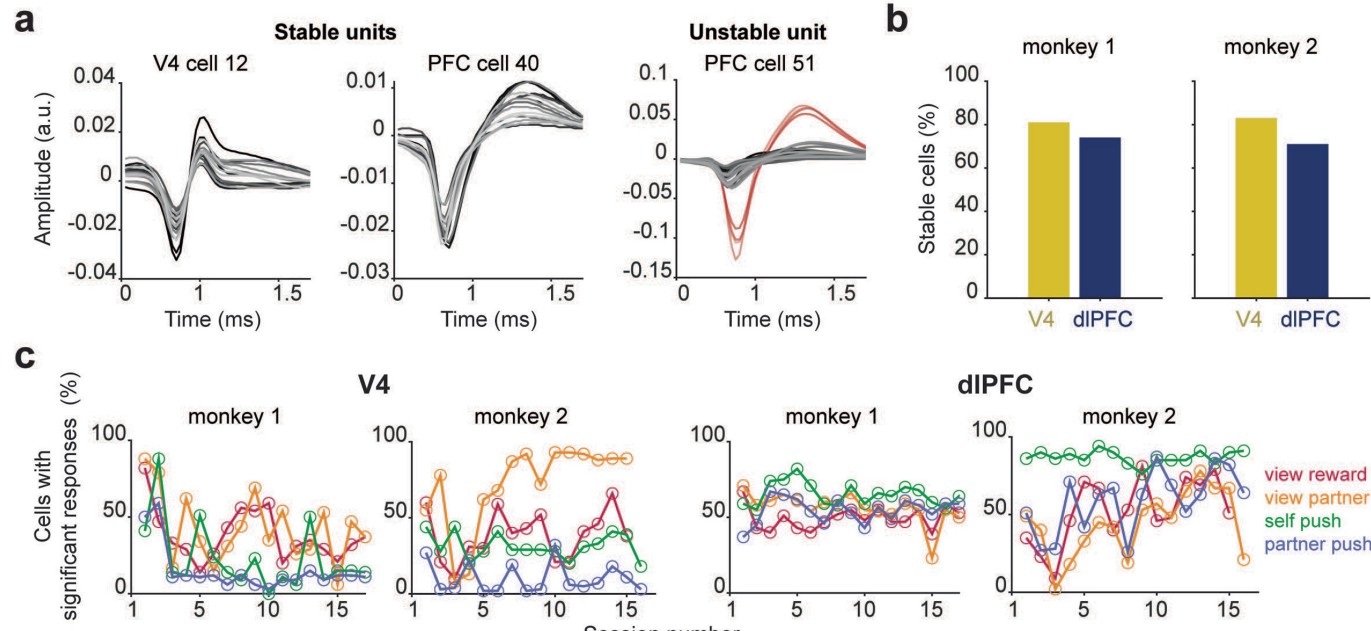

**Extended Data Fig. 3 | Neural population stability. a**, Example single units from one monkey showing spike waveforms recorded across sessions. Each panel represents the average waveform of the unit from one session, with session 1 plotted in a dark color and increasing in transparency across sessions. The unstable unit shows spike waveforms representing stable MUA (Black) and unstable SUA (red); the single unit was only present for 4 out of the 18 sessions. **b**, The number of stable cells divided by the total number of cells is the percentage of stable units in each area for each monkey. In monkey 1, 81% of recorded units (504/620) in V4 and 74% of recorded units in dlPFC (1350/1837) were consistent across sessions. In monkey 2, 83% of recorded units in V4 (1479/1773) and 71% of recorded units in dlPFC (561/794) were consistent. **c**, For each brain region, the percentage of cells out of the total recorded (M1: 34 V4 cells, 102 dlPFC cells; M2: 104 V4 cells, 46 dlPFC cells) that exhibited a statistically significant change in firing rate from baseline (intertrial interval firing rate) during social events (as shown in Fig. 3e but plotted across sessions for each monkey). For each cell, P < 0.01 Wilcoxon signed-rank test with FDR correction. The percentage of responding cells does not systematically change across sessions.

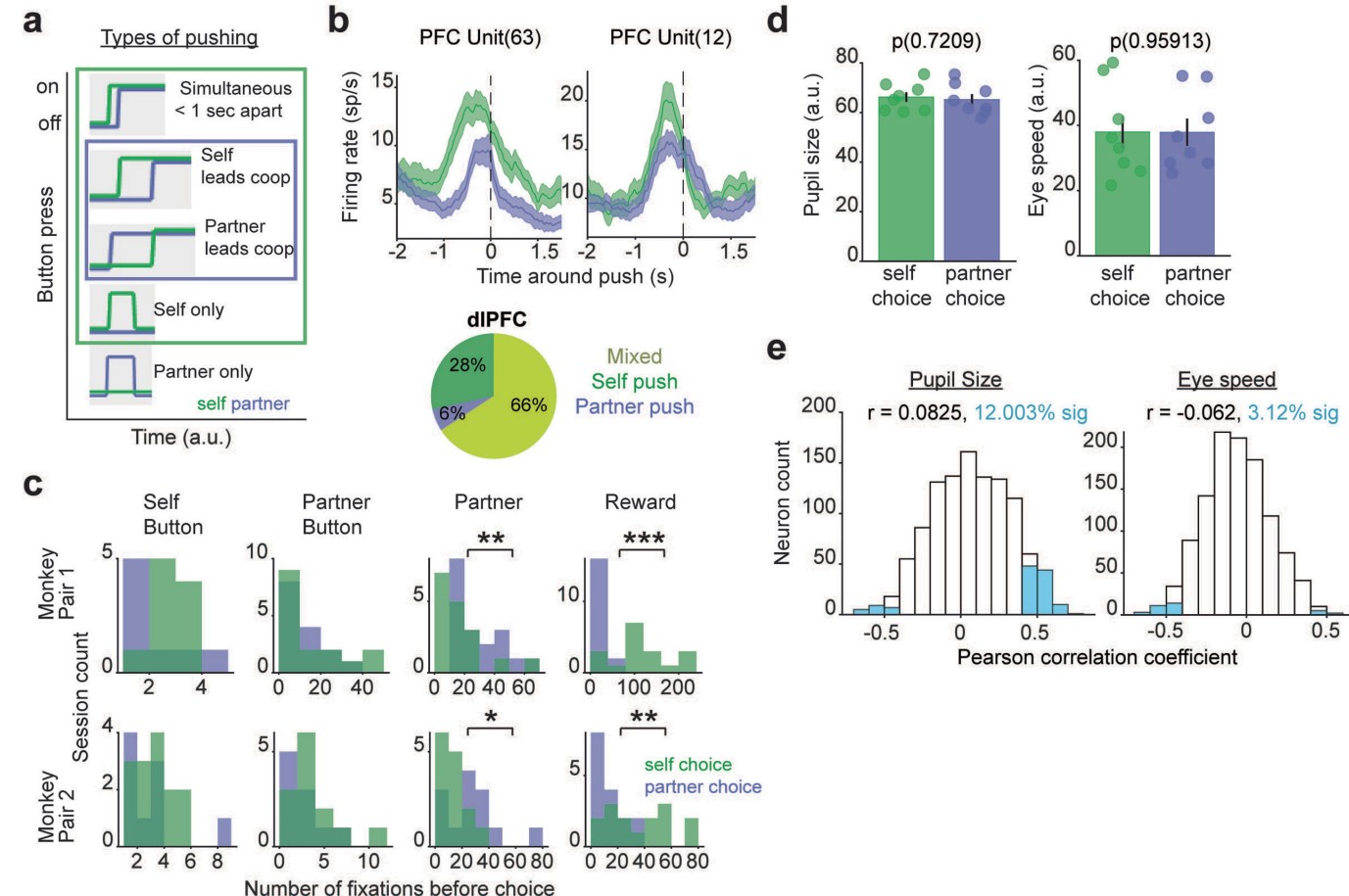

**Extended Data Fig. 4 | Neural responses and oculomotor events during pushes. a**, Self and partner pushes consist of push types that occurred in their respective outlined boxes. 'Partner only' pushes rarely occurred and were not used in analysis. For total number of pushes, see Methods: *Firing Rate and Response*. **b**, PSTHs from two example dlPFC units that show an increase in firing rate before self-monkey and partner pushes. Bottom: pie chart reflecting the percentage of push-modulated dlPFC units that respond only to self-push, only to partner push, or to both ("mixed"). Percentages averaged across sessions and monkeys. M1: 102 total dlPFC cells, 73 are push responsive; M2: 46 total dlPFC cells, 41 are push responsive. **c**, The distribution of the number fixations on each object that occurred before (1000 ms pre) self and partner (1000 ms pre, 500 ms post) pushes in each session. Self-monkey views the partner more during partner pushes compared to self-pushes, but he viewed the reward more before self-pushes. Pair 1 P Values: 0.005 and 5.79e$^{-5}$, Pair 2 P values: 0.03 and 0.003, Wilcoxon rank-sum test. **d**, Pupil size and eye speed, averaged across sessions and animals, that occurred before (1000 ms pre) the self and partner monkey pushes. There is no significant difference in pupil size and eye speed between animal's choices, Wilcoxon rank-sum test, P > 0.05. **e**, The distribution of Pearson correlation coefficients from the correlation of V4 and dlPFC neuron's firing rates with pupil size and eye speed occurring before (1000 ms pre) self and partner pushes. N = 1157 neurons from eight sessions across two animals. Percent significant represents neurons with a significant correlation coefficient, P < 0.01. *P < 0.05, **P < 0.01, ***P < 0.001.

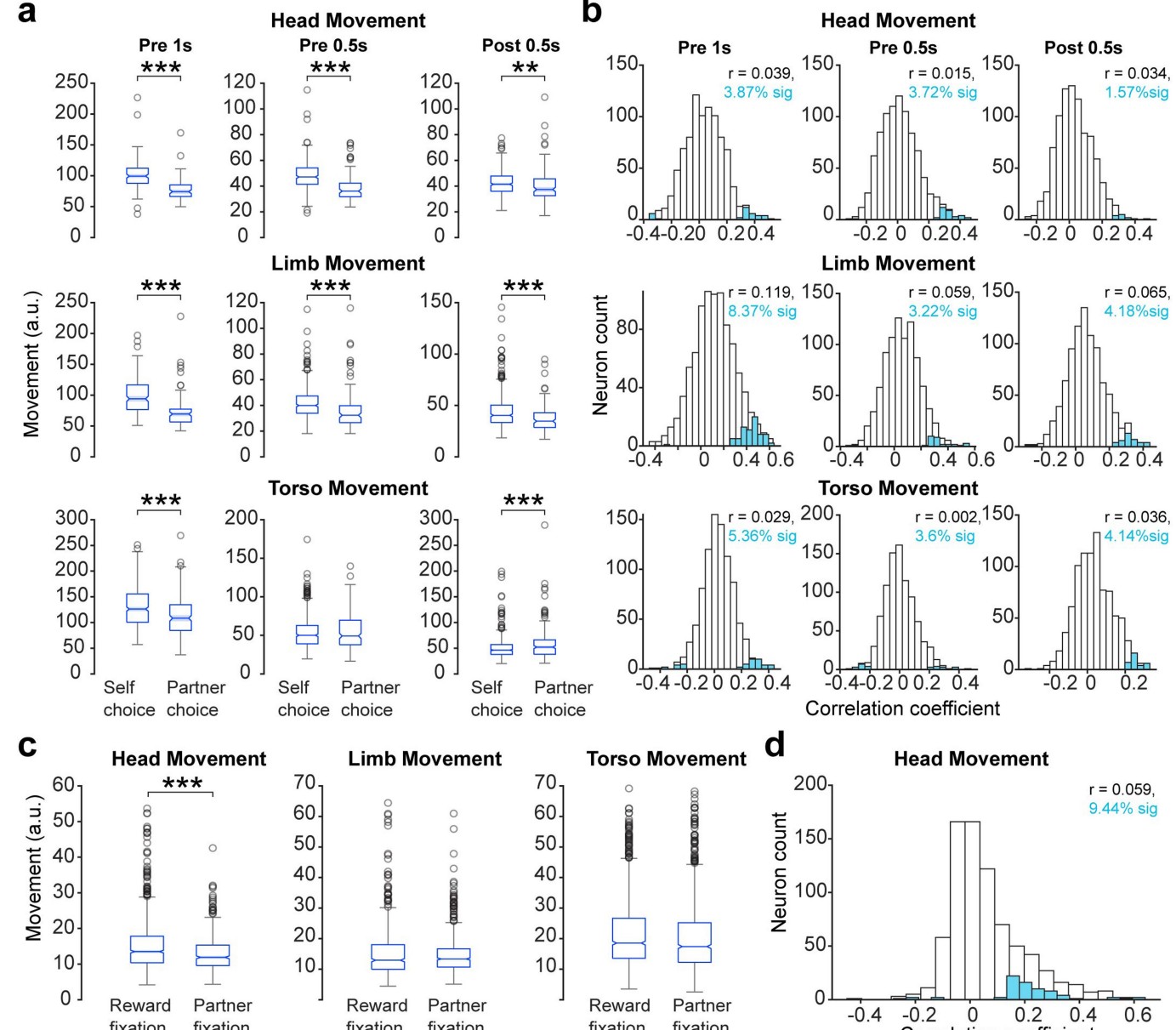

**Extended Data Fig. 5 | Neural firing rate correlations to movements during pushes and fixations. a**, Self-monkey's head movement, limb movement, or torso movement occurring around (1000 ms pre, 500 ms pre, or 500 post) self or partner monkey pushes, averaged across six sessions from two monkeys. Head movement: $P = 2.07e^{-19}$, $P = 2.49e^{-18}$, $P = 0.001$; Limb movement: $P = 7.12e^{-18}$, $P = 7.39e^{-11}$, $P = 2.49e^{-7}$; Torso movement: $P = 7.01e^{-9}$, $P = 0.46$, $P = 0.0007$; for Pre 1 s, Pre 0.5 s and Post 0.5 s respectively, Wilcoxon rank-sum test. On each boxplot, the central horizontal mark indicates the median, and the bottom and top edges of the box indicate the 25th and 75th percentiles, respectively. The whiskers extend to the most extreme data points not considered outliers, and the outliers are plotted individually using the 'o' symbol. **b**, Distribution of Pearson correlation coefficients from the correlation of V4 and dlPFC neuron's firing rates with head movement occurring around (1000 ms pre, 500 ms pre, or 500 post) self and partner pushes. N = 900 neurons from six sessions across two animals. "% sig" represents neurons with a significant correlation coefficient, $P < 0.01$. **c**, Self-monkey's head movement occurring 200 ms after onset of fixations on reward and partner monkey, averaged across six sessions from two monkeys. Head movement: $P = 2.44e^{-9}$; Limb movement: $P = 0.29$; Torso movement: $P = 0.0009$; Wilcoxon rank-sum test. While there is a significant difference in torso movement across reward and partner fixations, the magnitude of the difference is <2%. **d**, The distribution of Pearson correlation coefficients from the correlation of V4 and dlPFC neuron's firing rates with head movement occurring 200 ms after fixations on the reward system and partner monkey. N = 900 neurons from six sessions across two animals. "% sig" represents the % neurons with a significant correlation coefficient, $P < 0.01$. *$P < 0.05$, **$P < 0.01$, ***$P < 0.001$.

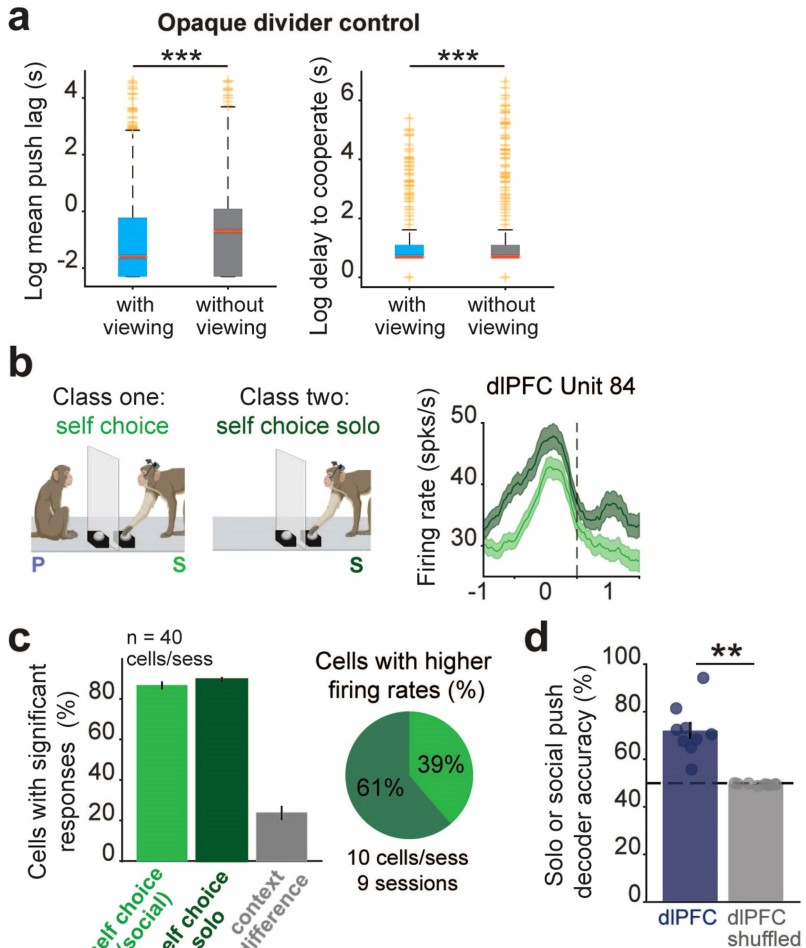

**Extended Data Fig. 6 | Non-social controls. a**, Left – Log of the average amount of time between self and partner monkey presses during learning ('with viewing') sessions and control sessions with the opaque divider ('without viewing'). P = 2.30e-08, Wilcoxon rank sum test. Right – Log of the average delay to cooperate, or time for both monkeys to be pressing from the start of a trial, during learning sessions and control sessions with the opaque divider. P = 1.078e-04, Wilcoxon rank sum test. Times were pooled across sessions (n = 4 sessions for each condition) and averaged across monkeys. On each boxplot, the central red mark indicates the median, and the bottom and top edges of the box indicate the 25th and 75th percentiles, respectively. The whiskers extend to the most extreme data points not considered outliers, and the outliers are plotted individually using the '+' symbol in gold. **b**, Social and solo trial schematic with a peri-event time histogram for a dlPFC cell that exhibits a significant change in firing rate between solo and social conditions, Wilcoxon rank-sum test, P < 0.05. **c**, Mean percentage of cells (n = 40 cells/ session from 9 sessions) responding significantly to self-choice in each condition when compared to baseline and compared across conditions (context difference), P < 0.01 Wilcoxon signed-rank test with FDR correction and Wilcoxon rank-sum test for context difference. Pie chart: Session averaged percentage of modulated (context difference) cells that exhibit significantly higher firing rates before self-choice during solo or social condition. **d**, Actual and shuffled decoding performance for solo and social trials using dlPFC activity occurring 1000 ms before self-choice, averaged across session values plotted as circles. P = 0.004, Wilcoxon signed-rank test. Dashed line represents chance. SEM is represented with error bars. *P < 0.05, **P < 0.01, ***P < 0.001. Illustrations in **b** were created using Biorender.

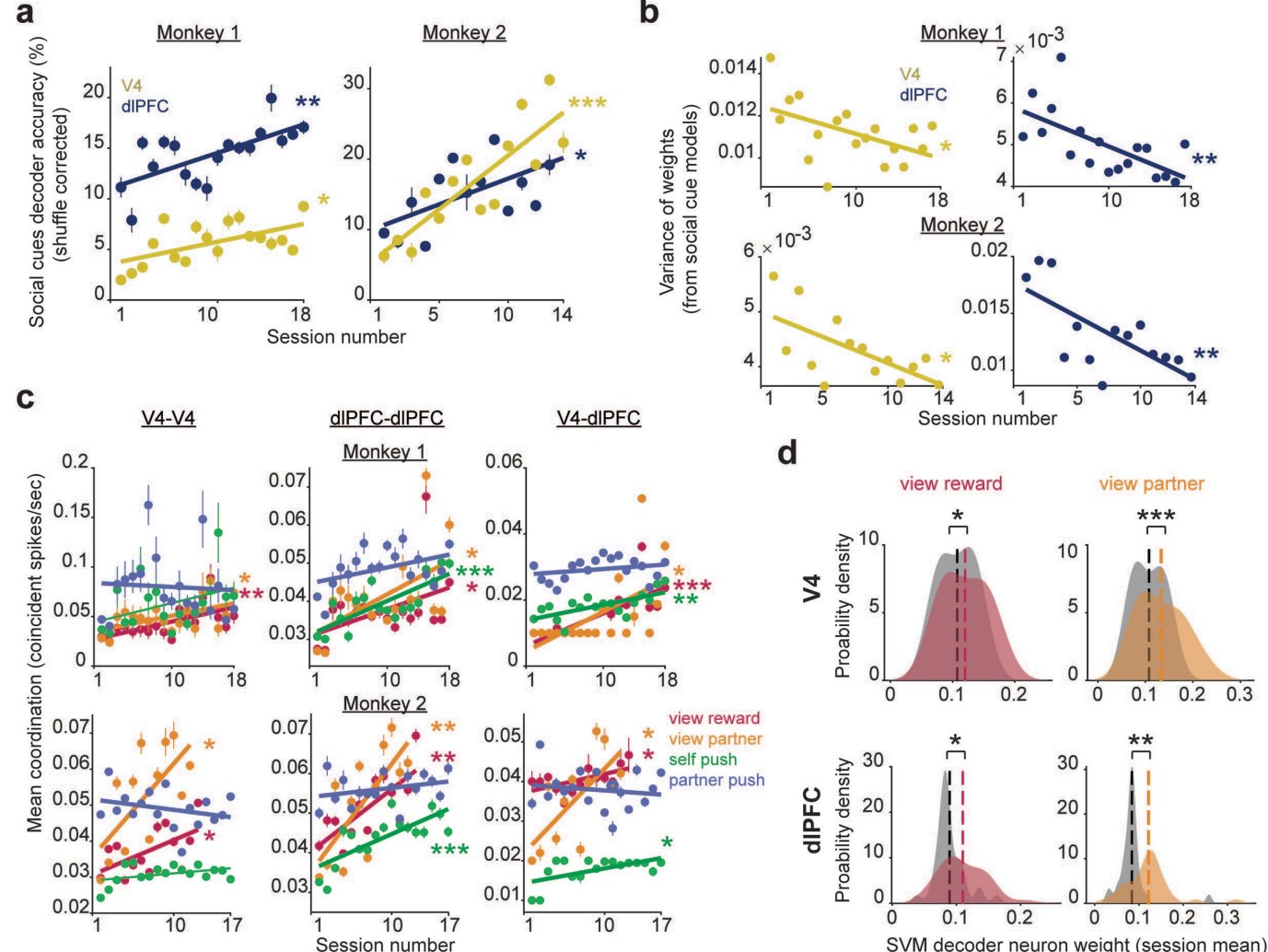

**Extended Data Fig. 7 | Neural correlates of learning cooperation from stable units only. a**, For each monkey, decoding accuracy for social cues from stable neural population activity in each brain area significantly improves during learning, as seen in Fig. 4a. V4 P = 0.01 and 1.32e⁻⁴, PFC P = 0.002 and 0.01; monkeys 1 and 2, linear regression. **b**, For each monkey, the variance of weights from the decoding models shown in panel 'a' significantly decreases across sessions during learning, as observed in Fig. 4g. V4 P = 0.03 and 0.01; PFC P = 0.004 and 0.005; monkey 1 and 2, linear regression. **c**, For each monkey, mean coordination of stable unit pairs for each social event in V4, dlPFC, and between brain areas is plotted across sessions. The same learning trends are observed as those shown in Fig. 5b, c and Extended Data Fig. 10a. Monkey 1

P-values: P = 0.007, 0.02, 0.09, 0.79; P = 0.03, 0.01, 1.93e⁻⁴, 0.07; P = 2.9e⁻⁴, 0.02, 0.002, 0.29; Monkey 2 P-values: P = 0.03, 0.01, 0.11, 0.26; P = 0.003, 0.006, 4.98e⁻⁴, 0.25; P = 0.03, 0.01, 0.01, 0.56; within V4, within PFC, and between areas respectively, linear regression. **d**, Probability density plots of decoder weights from stable, V4 and dlPFC correlated neurons during viewing social cues. Weights were averaged across neurons within each session for each monkey, then combined. Results are equivalent to those in Fig. 5e. V4 from left to right: P = 0.011 and P = 2.8e⁻⁴; PFC from left to right: P = 0.01 and 0.001, Wilcoxon signed-rank test comparing correlated neuron weights to remaining population. *P < 0.05, **P < 0.01, ***P < 0.001.

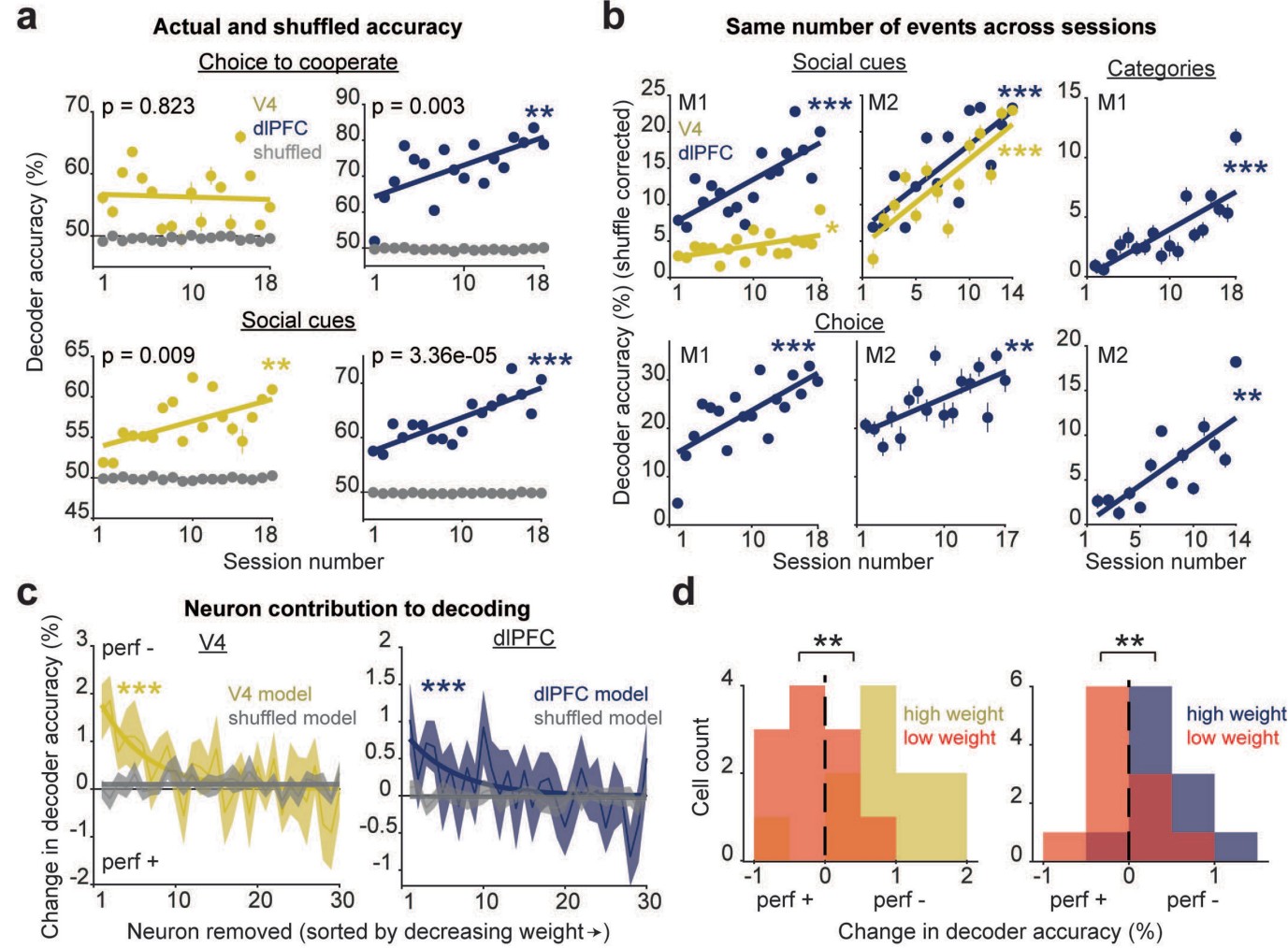

**Extended Data Fig. 8 | Decoding performance for social events. a**, Actual and shuffled decoding performance for each animal's choice to cooperate and discrimination of social cues. Actual and shuffled values are plotted to provide an example comparison for the shuffle-corrected plots completed for monkey 1, Fig. 4a and d. Shuffled decoder accuracies remained at chance levels (50%) across all sessions. This was also the case for every other decoding analyses in Fig. 4. **b**, Decoding performance for social cues, categories, and choice where the number of observations remained the same across all sessions and for each class. For each brain area, decoding accuracy still significantly improves during learning when the number of observations remains unchanged across sessions. All P-values are from linear regression and r is Pearson correlation coefficient. Social cues M1 dlPFC P = 0.0003, r = 0.75 and V4 P = 0.02, r = 0.53; M2 dlPFC P = 9.9e⁻⁴, r = 0.78 and V4 P = 0.0003, r = 0.83. Categories M1 dlPFC P = 1.3e-4, r = 0.78; M2 dlPFC P = 0.002, r = 0.76. Choice M1 dlPFC P = 6.84e-4, r = 0.72; M2 dlPFC P = 0.003, r = 0.68. **c**, The change in decoding performance

for social cues (original model accuracy with all neurons minus model with n-1 accuracy), is sorted according to the descending weight of the removed neuron. X-axis represents the index of a neuron; only one neuron was removed from each model. Session-averaged change in accuracy is plotted. Removing neurons with high weights decreases performance but the effect is attenuated as neurons with lower weights are removed. The change in accuracy for the first 30 neurons (out of 104 total in V4, 102 total in dlPFC) of descending weights is shown for clarity. V4 P = 1.11e-5, r = −0.71 and dlPFC P = 0.0009, r = −0.57; linear regression and Pearson correlation. **d**, For V4 and dlPFC, histograms display the change in decoding accuracy from removing upper and lower deciles of neurons (11 neurons) with the highest (gold and blue) and lowest (red) weights, respectively. Informative and uninformative neurons have significantly different effects on model performance. V4 P = 0.005 and dlPFC P = 0.009, Wilcoxon rank-sum test. *P < 0.05, **P < 0.01, ***P < 0.001.

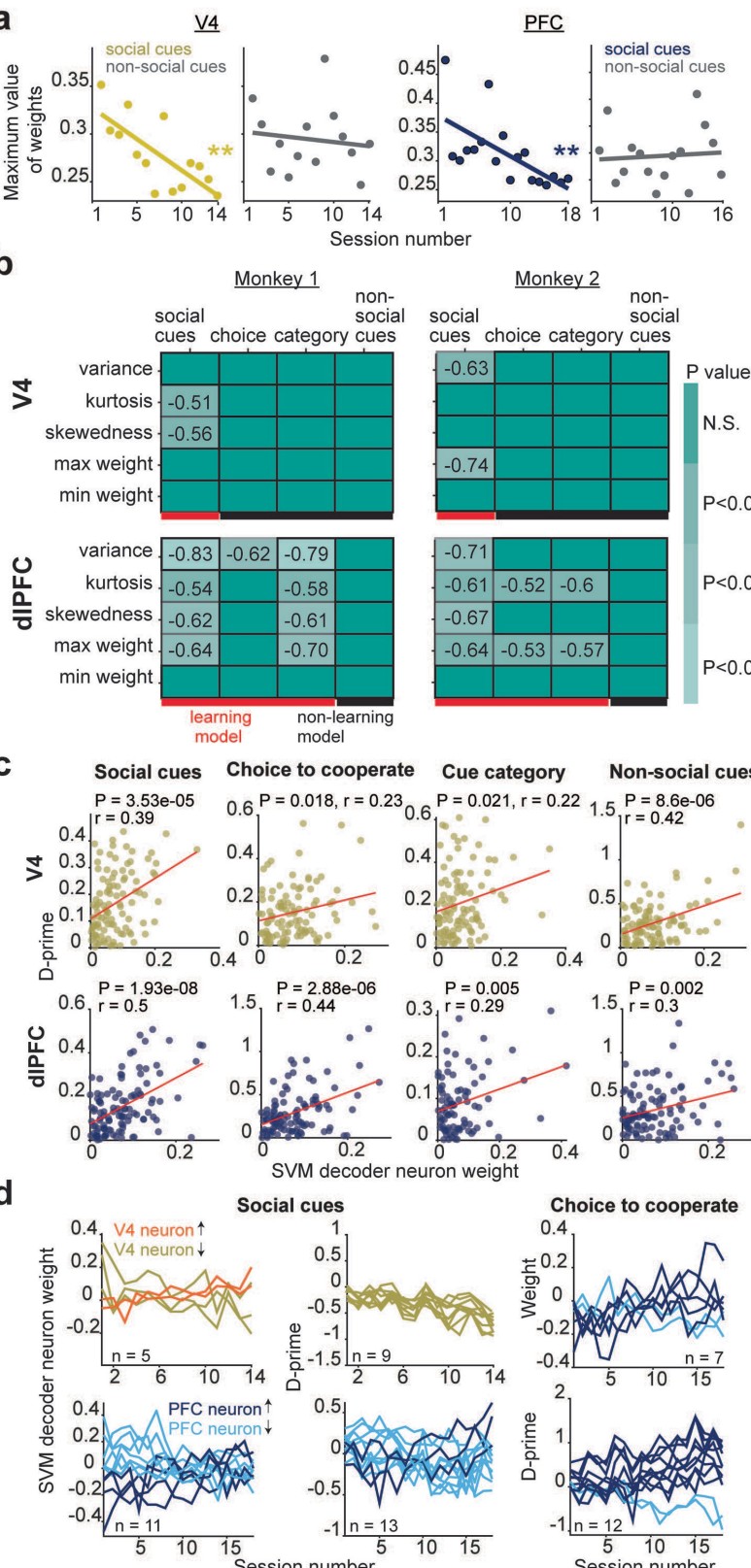

**Extended Data Fig. 9** | See next page for caption.

**Extended Data Fig. 9 | Learning reduces variance of neural population decoding weights. a**, The maximum absolute valued weight for each session in SVM models that decode social or non-social cues is plotted for each cortical area. V4 social cues maximum weight, P = 0.002, r = −0.74; non-social cues P = 0.65, r = −0.12; PFC social cues maximum weight, P = 0.004, r = −0.64; non-social cues P = 0.77, r = 0.07, linear regression and Pearson correlation. **b**, Summary of decoding models that exhibit decreased variance, kurtosis, skewedness, or maximum weight values for each brain area and monkey. For each decoding model, the P value, represented in shades of teal color, reflects linear regression of each weight metric with session number, as shown in panel a and Fig. 4g. Significantly decreased variance, kurtosis, skewedness, or maximum weight value is only observed in decoding models that exhibit increased decoding performance during learning. V4 P-values for monkey 1 kurtosis and skewedness P = 0.02 and P = 0.01, respectively. V4 P-values for monkey 2 variance and maximum weight P = 0.01 and 0.002, respectively. PFC P-values for monkey 1 variance, kurtosis, skewedness, and maximum weight values from social cues model are P = 1.67e⁻⁵, P = 0.03, P = 0.006, P = 0.004,

respectively; from choice model variance P = 0.005; from category model variance, kurtosis, skewedness, and maximum weight, P = 9.19e⁻⁵, P = 0.01, P = 0.006, P = 0.001, respectively. PFC P-values for monkey 2 variance, kurtosis, skewedness, and maximum weight values from social cues model are P = 0.004, P = 0.02, P = 0.008, P = 0.01, respectively; from choice model kurtosis and maximum weight, P = 0.02 and P = 0.03; from category model kurtosis, and maximum weight, P = 0.02 and P = 0.03, respectively. **c**, Within a session, neurons' decoding weight and D-prime values for task variables are positively correlated. Example sessions are shown for various decoding models where accuracy is above chance. Each circle represents the absolute value of D-prime and normalized SVM decoding weight of each neuron within a session. P-values and significant Pearson correlation coefficients are shown. **d**, For each cortical area, examples of individual neuron normalized weights and D-prime values that significantly increased (dark shade) or decreased (light shade) across sessions. N represents the total number of neurons that exhibited changes. In dlPFC, 75 stable neurons were recorded/session and in V4, 87 stable neurons were recorded/session. *P < 0.05, **P < 0.01, ***P < 0.001.

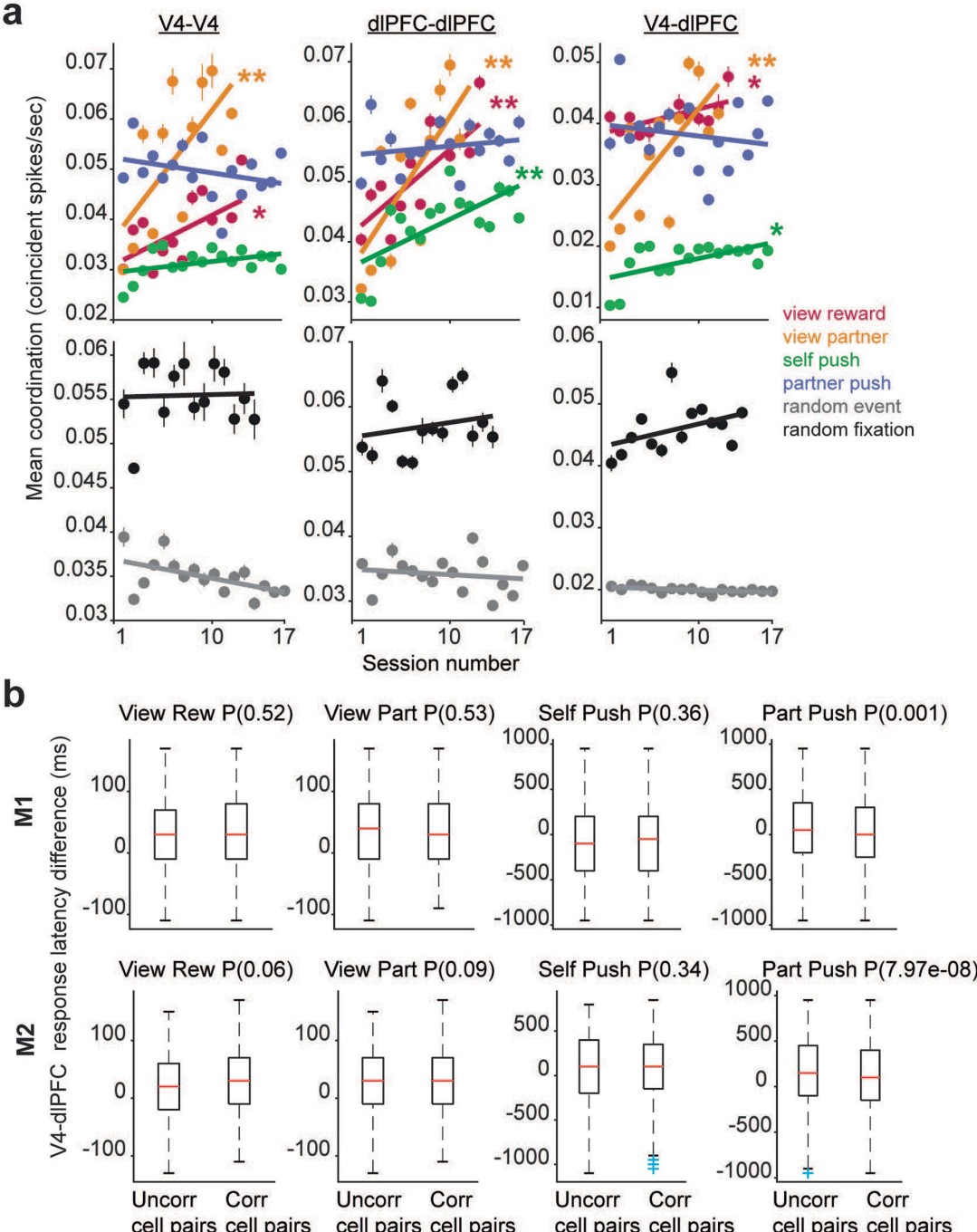

**Extended Data Fig. 10 | Spike-timing coordination and response latency.**
**a**, Top row: For Monkey 2, mean coordination plotted across sessions for each social event in V4, dlPFC, and between brain areas (V4: P = 0.03, r = 0.6; P = 0.005, r = 0.7; P = 0.1 and P = 0.2. PFC: P = 0.008, r = 0.7; P = 0.003, r = 0.8; P = 0.002, r = 0.7 and P = 0.44. V4-dlPFC: P = 0.02, r = 0.6; P = 0.006, r = 0.7; P = 0.01, r = 0.6 and P = 0.48). For 'view reward' and 'view partner' events, only 14 sessions were analyzed due to an inadequate number of stimulus fixations in 3 out of 17 sessions (sessions with <30 fixations were not included in the analysis). P-values for these data are reflected in Fig. 5c. Bottom row: For monkey 2, mean spike timing coordination during fixations on random objects and during random events (intertrial period, 4.5 seconds before trial start) for V4, dlPFC, and inter-areal cell pairs. V4: P = 0.03 and 0.9; PFC: P = 0.53 and 0.45; V4-dlPFC: P = 0.01 and 0.14, for random events and random fixations, respectively.

Significant P-values here correspond to decreasing trends. **b**, For each monkey (rows) and social event (columns), boxplots display the distribution of differences in V4 and dlPFC response latencies for correlated and uncorrelated neuron pairs across all sessions. V4 latencies were subtracted from dlPFC, i.e., negative values reflect pairs where the dlPFC neuron responded first. For uncorrelated pairs, the difference in latency between every possible combination of pairs was computed. The P-value from Wilcoxon rank-sum test comparing latency differences from correlated and uncorrelated pairs is displayed. On each boxplot, the central red mark indicates the median, and the bottom and top edges of the box indicate the 25th and 75th percentiles, respectively. The whiskers extend to the most extreme data points not considered outliers, and the outliers are plotted individually using the '+' symbol in blue.

# Reporting Summary

## Statistics

For all statistical analyses, confirm that the following items are present in the figure legend, table legend, main text, or Methods section.

| n/a | Confirmed | |
|---|---|---|
| ☐ | ☒ | The exact sample size (*n*) for each experimental group/condition, given as a discrete number and unit of measurement |
| ☐ | ☒ | A statement on whether measurements were taken from distinct samples or whether the same sample was measured repeatedly |
| ☐ | ☒ | The statistical test(s) used AND whether they are one- or two-sided *Only common tests should be described solely by name; describe more complex techniques in the Methods section.* |
| ☐ | ☒ | A description of all covariates tested |
| ☐ | ☒ | A description of any assumptions or corrections, such as tests of normality and adjustment for multiple comparisons |
| ☐ | ☒ | A full description of the statistical parameters including central tendency (e.g. means) or other basic estimates (e.g. regression coefficient) AND variation (e.g. standard deviation) or associated estimates of uncertainty (e.g. confidence intervals) |
| ☐ | ☒ | For null hypothesis testing, the test statistic (e.g. *F*, *t*, *r*) with confidence intervals, effect sizes, degrees of freedom and *P* value noted *Give P values as exact values whenever suitable.* |
| ☒ | ☐ | For Bayesian analysis, information on the choice of priors and Markov chain Monte Carlo settings |
| ☒ | ☐ | For hierarchical and complex designs, identification of the appropriate level for tests and full reporting of outcomes |
| ☐ | ☒ | Estimates of effect sizes (e.g. Cohen's *d*, Pearson's *r*), indicating how they were calculated |

*Our web collection on statistics for biologists contains articles on many of the points above.*

## Software and code

Policy information about availability of computer code

| | |
|---|---|
| Data collection | The neural data was collected using the the 'central' software (BlackRock Microsystems). The behavioral data and video data was collected using custom code in Matlab 2020b. The eye position and pupil size data was collected using the hardware and software provided by ISCAN Inc. |
| Data analysis | The neural data was pre-processed using Offline Sorter v4 (Plexon inc.). The behavioral, neural and part of the video material was processed using costum code in Matlab 2020b. The rest of the video material was processed using DeepLabCut (open source; version 2.0). |

For manuscripts utilizing custom algorithms or software that are central to the research but not yet described in published literature, software must be made available to editors and reviewers. We strongly encourage code deposition in a community repository (e.g. GitHub). See the Nature Portfolio guidelines for submitting code & software for further information.

## Data

Policy information about availability of data

All manuscripts must include a data availability statement. This statement should provide the following information, where applicable:
- Accession codes, unique identifiers, or web links for publicly available datasets
- A description of any restrictions on data availability
- For clinical datasets or third party data, please ensure that the statement adheres to our policy

All source data used to generate experimental Figures is available at https://zenodo.org/records/10384447. The raw data that support the findings of this study are available from the corresponding authors upon reasonable request.

## Research involving human participants, their data, or biological material

Policy information about studies with human participants or human data. See also policy information about sex, gender (identity/presentation), and sexual orientation and race, ethnicity and racism.

| | |
|---|---|
| Reporting on sex and gender | NA |
| Reporting on race, ethnicity, or other socially relevant groupings | NA |
| Population characteristics | NA |
| Recruitment | NA |
| Ethics oversight | NA |

Note that full information on the approval of the study protocol must also be provided in the manuscript.

# Field-specific reporting

Please select the one below that is the best fit for your research. If you are not sure, read the appropriate sections before making your selection.

☒ Life sciences          ☐ Behavioural & social sciences          ☐ Ecological, evolutionary & environmental sciences

For a reference copy of the document with all sections, see nature.com/documents/nr-reporting-summary-flat.pdf

# Life sciences study design

All studies must disclose on these points even when the disclosure is negative.

| | |
|---|---|
| Sample size | We conducted this cooperation study with two distinct monkey pairs consisting of three monkeys, but only recorded neural and eye tracking data from two monkeys in order to achieve robustness for the behavioral and electrophysiological results despite possible differences in the behavioral strategies between monkeys (all results were consistent across animals). We limited the number of neurally recorded monkeys to two for the main results to meet the requirements of lab animal use regulations that requires minimizing the number of animals in each study. Choosing the sample size of two is typical in electrophisiological studies in monkeys. |
| Data exclusions | For SVM modeling and cross-correlation, some sessions were excluded if they did not contain enough observations to perform the analysis. |
| Replication | Results were replicated across sessions and animals. The main experiments were repeated up to 20 times in each animal, and control experiments up to 10 times in each animal. |
| Randomization | Two distinct pairs of familiar animals completed the same cooperation experiment at separate times, with data always recorded from the subordinate animal in the pair. In control experiments, conditions were interleaved. |
| Blinding | Since no group allocation was done in this study, blinding was not required. |

# Reporting for specific materials, systems and methods

We require information from authors about some types of materials, experimental systems and methods used in many studies. Here, indicate whether each material, system or method listed is relevant to your study. If you are not sure if a list item applies to your research, read the appropriate section before selecting a response.

## Materials & experimental systems

| n/a | Involved in the study |
|-----|----------------------|
| ☒ | ☐ Antibodies |
| ☒ | ☐ Eukaryotic cell lines |
| ☒ | ☐ Palaeontology and archaeology |
| ☐ | ☒ Animals and other organisms |
| ☒ | ☐ Clinical data |
| ☒ | ☐ Dual use research of concern |
| ☒ | ☐ Plants |

## Methods

| n/a | Involved in the study |
|-----|----------------------|
| ☒ | ☐ ChIP-seq |
| ☒ | ☐ Flow cytometry |
| ☒ | ☐ MRI-based neuroimaging |

# Animals and other research organisms

Policy information about studies involving animals; ARRIVE guidelines recommended for reporting animal research, and Sex and Gender in Research

| | |
|---|---|
| Laboratory animals | Three adult male rhesus monkeys (Macaca mulatta) were used in this study, including Monkey M age 11, Monkey R ages 10 and 12, and Monkey G age 16 during the time of experiments. |
| Wild animals | The study did not involve wild animals. |
| Reporting on sex | Due to the limited number of animals used for the main study (n=3), no reporting on sex or gender was possible. |
| Field-collected samples | The study did not include field-collected samples |
| Ethics oversight | All experiments were performed in accordance with The Animal Welfare Committee (AWC) and the Institutional Animal Care and Use Committee (IACUC) for McGovern Medical School at The University of Texas Health Science Center at Houston. |

Note that full information on the approval of the study protocol must also be provided in the manuscript.

# Plants

| | |
|---|---|
| Seed stocks | *Report on the source of all seed stocks or other plant material used. If applicable, state the seed stock centre and catalogue number. If plant specimens were collected from the field, describe the collection location, date and sampling procedures.* |
| Novel plant genotypes | *Describe the methods by which all novel plant genotypes were produced. This includes those generated by transgenic approaches, gene editing, chemical/radiation-based mutagenesis and hybridization. For transgenic lines, describe the transformation method, the number of independent lines analyzed and the generation upon which experiments were performed. For gene-edited lines, describe the editor used, the endogenous sequence targeted for editing, the targeting guide RNA sequence (if applicable) and how the editor was applied.* |
| Authentication | *Describe any authentication procedures for each seed stock used or novel genotype generated. Describe any experiments used to assess the effect of a mutation and, where applicable, how potential secondary effects (e.g. second site T-DNA insertions, mosiacism, off-target gene editing) were examined.* |

