## [Peer Review File · Nature]

Manuscript Title: Visuo-frontal interactions during social learning in freely moving macaques

Reviewer Comments & Author Rebuttals

Reviewer Reports on the Initial Version:

Referees' comments:

Referee #1 (Remarks to the Author):

This paper reports a study of the neural population computations and the neuron-to-neuron interactions between visual and frontal areas during learning of social cooperation. The study is technically impressive and the results are truly exciting.

The main claims seem to be that 1) As monkeys learn social cooperation, they redistribute socially-relevant information among neurons within each area. 2) Decoding population activity revealed that viewing social cues influences the decision to cooperate. 3) Learning social events increased coordinated spiking between visual and prefrontal cortical neurons, which was associated with improved accuracy of neural populations to encode social cues and the decision to cooperate.

The novelty and significance of the paper relies on the solidity and credibility of the above claims. While I found the analyses of the authors compelling, I feel that more work is needed to strongly prove and better qualify the above statements. I detail my concerns below, and in some cases I offer some suggestions on how to address them. While I have confidence that the new analyses will still support the claims of the authors, I still think it is necessary to better prove the claims to make the paper even more exciting. It is possible that some of my concerns are already addressed by authors in extra analyses provided as supplement that I may have missed or misunderstood. When that's the case, it is enough that the authors describe them more prominently or clearly in revision. But my impression is that several of the concerns below would require additional control analyses.

The statement about improved information encoding during learning (or following visual cues) all rely on SVM decoders of neural population activity. The authors perform their analyses with care and competence, for example balancing the data between conditions to be decoded when constructing the decoder. However, the performance of the decoder does not only depend on the information in the data, but also on the number and nature of data available for training. For example, having less trials of a certain type may be detrimental for training the weights of the SVM (if the authors run the same SVM but throwing away randomly half of the data it will presumably perform less well because training is worse. On the other hand, reducing data non-randomly (because of data selection) may make data more homogeneous and artificially push up decoding performance. Across learning or across analyses, the numbers of events that goes into the classifier seems to change. Also, in some analyses (like on page 8) some decoding analyses are done after data exclusion (e.g. excluding button pushes preceded by fixation on social clues..). I could not find in the paper a sufficiently strong discussion or control analyses for these potential issues. Do they affect

the conclusions? Can the authors conceive for example data-matched analyses that can rule out some of these possible confounders?

Decrease in variance of normalized SVM weights across learning is taken as evidence that the population distributes information across more neurons as learning goes up. The authors argue that this distribution is the key computation taking place in each area across learning. I am not yet entirely convinced about this conclusion. The normalized analysis may mask the contribution of increased averaged single neuron information. Also the normalized weights do not necessarily reflect correctly the single-trial information that the neurons contribute (this would be quantified e.g. by the performance of the single-neuron decoder). The method of ref 35 is not the established gold standard in the field. A more principled and established method would be running decoders and population analyses with neurons progressively added or removed. Also, it would be useful to have information decoded for single neurons, to understand whether an increase of average information per neuron is also present. Population scaling analyses like the one described above, which are more standard in population coding analysis, would have the potential to help the authors further refining their description of population computations during learning of social cues, and to refine their conclusions about it. I do not expect that their main conclusion of higher distribution would change, but it is reasonable to expect that the new analyses would add new interesting results.

In the correlation analyses, the authors do not find major changes in the within-area neuron-to-neuron coordination. However, they found a very interesting increase in the time-lagged visual-prefrontal coordination especially between informative neurons. The result is very exciting, but it is unclear to me whether the higher coordination between informative neuron pairs can to some extent be explained just by the fact that the informative individual neurons have activity more precisely time locked to the external event that they respond to, without a real increase of communication between the neurons? Did the authors control for this possibility? If not, can they control for it in further analyses?

The increase in correlations between neurons is interpreted as “Temporal coordination of neuronal spiking is believed to be correlated with signal transmission” citing ref 36. On this I have two comments. First, most of the coordination increase seems across areas and with longish time lags. This seems more related to enhancing in some way some aspects of the correlation code (e.g. making it more consistent and so easier to be read out across areas (e.g. Runayn et al Nature 207, Valente et al Nature Neurosci 2021) than to improving signal propagation due to coincidence detection, as the current sentence seems to imply. I feel that the relevance of the coordinated activity found by the authors could be better discussed.

Second, original references such as the older papers of Christoph Koch or of Emilio Salinas seem more suitable than ref 36 to support the current sentence.

Minor: Page 19: eq for normalized weights seems wrong (square root bar ends too soon..)

Referee #3 (Remarks to the Author):

This manuscript presents the results from an ambitious and technical tour de force study that aimed to investigate the neural mechanisms underlying social cooperation in freely moving macaques. The researchers wirelessly recorded the spiking activity of neurons in one visual (V4) and one prefrontal (dlPFC) cortical areas while pairs of macaques engaged in a task that required social cooperation (i.e., pushing buttons together to obtain rewards). The authors also wirelessly tracked the eye positions of the recorded monkeys (1 of the 2 monkeys involved in the task for a given pair). They found that viewing social cues facilitated the decision to cooperate, and that learning socially contingent events increased coordinated spiking between visual and prefrontal cortical neurons. These findings suggest that the visuo-frontal cortical network prioritizes relevant sensory information to facilitate learning social interactions (behaviors underlying a situation where two monkeys both have to respond for obtaining rewards) while freely moving macaques interact in a naturalistic environment.

Overall, this manuscript is very interesting in terms of the task design, behavioral and neural data, as well as the data analysis. Their task design is very novel, and their methods, including using the wireless neural recording and eye tracking, can be very powerful for future studies and other researchers. The results are robust and clear. The authors also did an excellent job writing the article and presenting the figures intuitively and informatively. Generally, I am very glad to see these results (after seeing this exciting presentation at the SfN meeting). However, there are a few issues that require some attention, and it would be great if the authors could address them. Here are the major comments:

1. I am wondering whether it's possible to separate the "fixations on partners" into more detailed variables, such as "fixations on head" or "fixations on body" (relevant to Fig 2a). I am very curious to know, in the later session analysis, the transition probability corresponding to "view partner -> push" and "push -> view partner" are related to viewing different parts of the partner. Could it be "view partner's head -> push"? Given that the authors used multi-point pose tracking, this can be tested and might provide more nuanced information on how they were using the social cues.

2. It is unclear to me why the transition probability between "self push" and "partner push" was low, and did not increase with training? (relevant to Fig 2e). I would expect that this transition probability to also increase with training, because essentially it's reflecting the cooperation (two animals pushing almost at the same time) to become more efficient at obtaining rewards. Maybe it's due to the time bin size the authors used to create the behavioral event time series. I wonder how would this look if one uses a larger time bin that might fit better with a longer timescale. From Fig 1e, it seems that there was a 1-2 s gap between the two pushes, so maybe using a time bin that reflects this time gap might be more appropriate or at least informative.

3. I was not sure why the number of green bars is smaller than the purple bars in the top panel of Fig. 3c (based on x axis values), but they seem to be higher in the bottom line-plot panel (based on

the y axis values).

4. If I understand it correctly, the major difference between “fixation on social cue” and “fixation on partner” is that social cue also included reward. Could the authors clarify what types of “view reward” was included as the social cue? Did it only include the rewarding event or the reward itself of the partner monkey? Going back to Fig. 3c, does the difference basically indicate that there were a lot more pushes with fixation on reward in “self choice” than in the “partner choice”?

5. Spike cross-correlation should be sensitive to the neurons’ firing rate (higher firing rates in at least one of the neuron of the pair relate to high cross-correlation), did the authors address this confound in their analysis? I am wondering whether the higher spike correlations during the behavior events were largely due to the higher firing rate during the same time window. To test the directionality of the communication between the brain areas, it’s common to analyze the spike-field coherence. Did the authors have tried analyzing the spike-field relationship or some directionality analysis using LFP (such as granger or PDC, or DTF)? Ideally, it should reveal the similar finding as presented here, and even add a new dimension (i.e., frequency).

6. About Fig. 5d, there seems to be in general higher number of “V4 leads” than “dIPFC leads” but there are a plenty of cells showing the opposite pattern as well (the differences are not as one-sided) and thus it’s unclear if these proportion differences are significant.

7. One of the concerns that drew a lot of my attention is that there’s no real non-social control in this task. Ideally, if one wants to claim that animals were doing the task in a social and cooperative manner, it’s better to do controls such as blocking the view of seeing the other animal, two animals pushing without cooperative contingency, or fake or computer partner instead of a real monkey. However, after reading this manuscript, I am generally convinced that the two animals WERE cooperating. I believe that the learning paradigm really helped here – animals had an increased level of coincident pushes over training, and the transition probability between social events and push also increased. It could be better if the authors also had a control showing that if there is no need to cooperate, there will be no social view to push transition. But I understand that the authors did not particularly claim anything about social specificity of the findings – in fact, the findings themselves are very interesting still given the naturalistic paradigm and environmental contingencies involving another conspecific. Also, I can see that it would be hard to have such controls, especially the authors observed and reported the learning phase – e.g., once the animals learned, it’s hard to have a pure control because they cannot unlearn what they learned and they may still use the social strategy to do the non-social controls. In addition, it is not too surprising that dIPFC and V4 neurons generally did not show much agent specificity given that these areas prioritize sensory and motor aspects of complex behaviors (as opposed to, say, medial areas). For example, Fig. 4f being a “social learning model” or more generic “learning model” in a complex environment remains unclear. Regardless, I would like to see the authors’ comments about the missing controls in Discussion to guide the readers on the interpretation. Just to be clear, I am NOT recommending adding any non-social control here.

8. The authors provided evidence that neural selectivity changes over the course of learning. The interpretation is that the coding scheme is shifting more toward a distributed coding scheme.

However, it is less clear if the coding is becoming more distributed per se or the coding is being degraded over time, in a sense of representational decay, or changing over learning, in a sense that the information is more allocated toward some downstream areas. Do authors propose the distributed coding scheme based on the decrease in weight distribution specifically for social cues where the decoding performance increased? I wonder if this specific combination of results could also happen when neurons begin to be more and more de-correlated to signal one (or just few) specific events in the behavioral environment over the course of the session (as opposed to more and more neurons are beginning to represent the shared information as in a distributed coding scheme)?

Reviewer 1

1. The statement about improved information encoding during learning (or following visual cues) all rely on SVM decoders of neural population activity. The authors perform their analyses with care and competence, for example balancing the data between conditions to be decoded when constructing the decoder. However, the performance of the decoder does not only depend on the information in the data, but also on the number and nature of data available for training. For example, having less trials of a certain type may be detrimental for training the weights of the SVM (if the authors run the same SVM but throwing away randomly half of the data it will presumably perform less well because training is worse). On the other hand, reducing data non-randomly (because of data selection) may make data more homogeneous and artificially push up decoding performance.

1.1 Across learning or across analyses, the numbers of events that goes into the classifier seems to change.

We definitely agree with the general point made by the reviewer. In our study, the number of events does indeed change across sessions, but this did not influence our conclusions. Animals cooperated more quickly as they learned and sessions became shorter, so the number of events (fixations and pushes) typically decreased across sessions. For instance, on average (across sessions and monkeys), the number of fixations decreased by 95% and the number of pushes decreased by 50%. One might expect that this reduction in the number of observations across sessions would result in a decrease in decoding performance, but in fact, we found the opposite. To verify this, we conducted the classification analysis in Figure 4a, and 4c-d, where we balanced the number of observations both between classes and across sessions, as urged by the reviewer. For each monkey and brain area, decoding accuracy still significantly improved during learning when the number of observations remains the same across sessions. We actually performed this analysis and checked this ourselves before initial submission, and thank the reviewer for drawing attention to its relevancy. We included the results below and in Extended Data Figure 10b, and updated the Methods section to reflect this new analysis.

Extended Data Figure 10b. b, Decoding performance for social cues, categories, and choice where the number of observations remained the same across all sessions and for each class. For each brain area, decoding accuracy still significantly improves during learning when the number of observations remains

unchanged across sessions. Social cues M1 dlPFC: $P = 0.0003$, $r = 0.75$, and V4: $P = 0.02$, $r = 0.53$; M2 dlPFC: $P = 9.9 \times 10^{-4}$, $r = 0.78$, and V4: $P = 0.0003$, $r = 0.83$ for linear regression and Pearson correlation. Categories M1 dlPFC: $P = 1.3 \times 10^{-4}$, $r = 0.78$; M2 dlPFC: $P = 0.002$, $r = 0.76$ for linear regression and Pearson correlation. Choice M1 dlPFC: $P = 6.84 \times 10^{-4}$, $r = 0.72$; M2 dlPFC: $P = 0.003$, $r = 0.68$ for linear regression and Pearson correlation. * $P < 0.05$, ** $P < 0.01$, *** $P < 0.001$, linear regression.

1.2 Also, in some analyses (like on page 8) some decoding analyses are done after data exclusion (e.g. excluding button pushes preceded by fixation on social clues..). I could not find in the paper a sufficiently strong discussion or control analyses for these potential issues. Do they affect the conclusions? Can the authors conceive for example data-matched analyses that can rule out some of these possible confounders?

The reviewer mentions data exclusion in the analysis in Figure 4e, discussed on page 6. In Methods, page 19, we stated that “the number of observations was matched between “with cue” and “without cue” classes to enable a fair comparison of decoder performance across conditions.” Therefore, changes in decoding performance are not due to one condition having fewer observations, or data, than the other. However, if the reviewers and the editor deem it necessary, we would be happy to include additional text or update the Methods section to further clarify this issue besides what is stated already in the Methods section.

2. Decrease in variance of normalized SVM weights across learning is taken as evidence that the population distributes information across more neurons as learning goes up. The authors argue that this distribution is the key computation taking place in each area across learning. I am not yet entirely convinced about this conclusion.

2.1 The normalized analysis may mask the contribution of increased averaged single neuron information. Also the normalized weights do not necessarily reflect correctly the single-trial information that the neurons contribute (this would be quantified e.g. by the performance of the single-neuron decoder). The method of ref 35 is not the established gold standard in the field.

We appreciate reviewer’s concern about weight normalization. Because the weight distributions from two models or different sessions can be scaled differently (even if models are trained on the same data set), weights need to be normalized within a session in order to rigorously compare them across sessions. During normalization, each neuron’s weight was divided by the square root of the sum of squares of all other weights in the population. Therefore, if a neuron had a ‘higher’ weight relative to the rest of the population before normalization, it will still have a ‘higher’ weight after normalization. The paper we cited (Koren et al., 2021) arguably reflects the standardization of this methodology in neuroscience, but if the reviewer recommends additional literature, we can include it. Importantly, computing the variance and other metrics displayed in Extended Data Figure 9 with non-normalized weights yields similar results.

2.2 A more principled and established method would be running decoders and population analyses with neurons progressively added or removed.

As suggested by the reviewer, we repeated the decoding analysis for social cues and removed one neuron at a time according to its normalized weight in the model built from the entire population of neurons. Each decoding model used $n-1$ neurons, where n is the entire population of cells. For instance, the change in decoder performance corresponding to neuron ‘10’ represents the model containing $n-1$ cells, with only the 10th best cell (i.e., sorted based on the weights) being removed. Removing neurons from decoders likely results in decreased performance, regardless of the weight. However, we found that removing neurons with the highest weight had the greatest impact in decoding performance, resulting in the largest decreased performance when compared to the decoder using all the neurons (Extended Data Fig. 10c). As shown below, in each cortical area, there is a significant decrease in performance across neurons of

descending weight value. Shuffled model performance was unaffected by the removal of any neuron (Extended Data Fig. 10c). Finally, removing neurons with the highest weights significantly decreased decoding performance, but removing neurons with the lowest weights slightly improved performance (Extended Data Fig. 10d). These findings were observed in each session, and systematic changes in these metrics did not occur across sessions during learning. We added these results to Extended Data Figure 10. Overall, this analysis reveals that neurons with higher weights contribute more to encoding, and this does not change during learning.

Extended Data Figure 10c-d. **c**, Change in decoding performance for social cues (original model accuracy with all neurons minus model with n-1 accuracy), sorted according to the descending weight of the removed neuron. X-axis represents the index of a neuron; only one neuron was removed from each model. Session-averaged change in accuracy is plotted. Removing the neurons with high weights decreases performance but the effect is attenuated as neurons with lower weights are removed. The change in accuracy for the first 30 neurons (out of 104 total in V4, 102 total in dIPFC) of descending weights is shown for clarity. V4 P-value = 1.11×10^{-5} , $r = -0.71$ and dIPFC P-value = 0.0009, $r = -0.57$; linear regression and Pearson correlation. **d**, For V4 and dIPFC, histograms display the change in decoding accuracy after removing upper and lower deciles of neurons (11 neurons) with the highest (gold and blue) and lowest (red) weights, respectively. Informative and uninformative neurons have significantly different effects on model performance. V4: P = 0.005; dIPFC: P = 0.009, Wilcoxon rank-sum test.

2.3 Also, it would be useful to have information decoded for single neurons, to understand whether an increase of average information per neuron is also present. Population scaling analyses like the one described above, which are more standard in population coding analysis, would have the potential to help the authors further refining their description of population computations during learning of social cues, and to refine their conclusions about it. I do not expect that their main conclusion of higher distribution would change, but it is reasonable to expect that the new analyses would add new interesting results.

We directly addressed reviewer's concern. For each neuron, we computed its D-prime value for distinguishing between self and partner push, fixations on social cues (reward system and partner), fixations on non-social cues (floor and self-button), and social and non-social category fixations. Neurons' D-prime values were positively correlated with the weight in the decoding models (Extended Data Figure 9c). This finding was true across animals, cortical areas, and all decoding models where performance was above chance. Additionally, some individual neurons (~15%) exhibited systematic changes in D-prime or decoding weights during learning (Extended Data Fig. 9d), but we did not observe any systematic changes in correlations between D-prime values and neural weights across sessions. Therefore, during learning, an

individual neuron's sensitivity to an event can develop across sessions, and these changes could support the improved decoding performance seen at the population level.

Extended Data Figure 9c-d. **c**, Within a session, neurons' decoding weight and D-prime values for task variables are positively, and significantly, correlated. Example sessions are shown for various decoding models where accuracy is above chance. Each circle represents the absolute value of D-prime and normalized SVM decoding weight of each neuron within a session. P-values and significant correlation coefficients are shown. **d**, For each cortical area, examples of individual neuron normalized weights and D-prime values that significantly increased (dark shade) or decreased (light shade) across sessions. N represents the total number of neurons that exhibited changes. In dIPFC, 75 stable neurons were recorded/session and in V4, 87 stable neurons were recorded/session.

3. In the correlation analyses, the authors do not find major changes in the within-area neuron-to-neuron coordination. However, they found a very interesting increase in the time-lagged visual-prefrontal coordination especially between informative neurons. The result is very exciting, but it is unclear to me whether the higher coordination between informative neuron pairs can to some extent be explained just by the fact that the informative individual neurons have activity more precisely time locked to the external event that they respond to, without a real increase of communication between the neurons? Did the authors control for this possibility? If not, can they control for it in further analyses?

This is an important control analysis requested by the reviewer. We clarify that within brain areas, we also see an increase in spike timing coordination (e.g., at ± 6 ms lag) for some of the social events (Figure 5b

and Extended Data Figure 11). Regarding reviewer's question about increased coordination between inter-areal correlations being caused by more precise time-locking of neural responses to social events, we performed two analyses to address this. First, we analyzed the latency of firing rates relative to event onset for each V4-dIPFC correlated neurons (i.e., informative neurons), and for the neurons that were not significantly correlated (uninformative neurons). For each cell, we defined response latency as the time taken by firing rates to reach half maximum rate from event onset. Our analysis revealed that neurons' response latency did not significantly or systematically change across sessions, i.e., neurons did not become more precisely time locked to the events. This was true in each cortical area of each monkey, both for the correlated/informative neuron pairs as well as the entire population.

Second, we analyzed the difference in response latency between correlated V4-dIPFC neurons and uncorrelated V4-dIPFC neuron pairs. We found that the distributions of pairwise latencies between correlated neurons and the remaining, uncorrelated population, were not significantly different (Extended Data Fig. 11b). This was true for each social event except partner choice, which did not exhibit improved spike-timing coordination during learning. We added these findings to Extended Data Figure 11. Altogether, these results indicate that improved spike timing coordination between informative, correlated, V4-dIPFC neurons is not caused by neuronal activity that is more precisely time locked to external events, but instead is due to shared interaction or communication between them.

Extended Data Figure 11b. b. For each monkey (rows) and social event (columns), boxplots display the distribution of differences in V4 and dIPFC response latencies for correlated and uncorrelated neuron pairs across all sessions. We subtracted V4 latencies from those in dIPFC, i.e., negative values reflect pairs where the dIPFC neuron responded first. For uncorrelated pairs, the difference in latency between every possible combination of pairs was computed. The P-value from Wilcoxon rank-sum test comparing latency differences from correlated and uncorrelated pairs is displayed at the top of each plot. On each box, the central red mark indicates the median, and the bottom and top edges of the box indicate the 25th and 75th percentiles, respectively. The whiskers extend to the most extreme data points not considered outliers, and the outliers are plotted individually using the '+' marker symbol in blue.

4. The increase in correlations between neurons is interpreted as “Temporal coordination of neuronal spiking is believed to be correlated with signal transmission” citing ref 36. On this I have two comments. First, most of the coordination increase seems across areas and with longish time lags. This seems more related to enhancing in some way some aspects of the correlation code (e.g. making it more consistent and so easier to be read out across areas (e.g. Runyan et al Nature 2007, Valente et al Nature Neurosci 2021) than to improving signal propagation due to coincidence detection, as the current sentence seems to imply. I feel that the relevance of the coordinated activity found by the authors could be better discussed. Second, original references such as the older papers of Christoph Koch or of Emilio Salinas seem more suitable than ref 36 to support the current sentence.

We agree with the reviewer that our interpretation of spike-timing coordination could be improved. Specifically, our finding of improved spiking coordination between areas during learning could reflect an increase in synaptic strengths between cells and/or improved population coupling (i.e., communication subspace, Semedo et al., Neuron 2019) between regions. These possibilities are not mutually exclusive, and both could facilitate the improved spike timing coordination that we observed during learning. We have now updated the discussion and the “Spike-timing coordination increases during learning cooperation” section of the manuscript, pages 10 and 12. We also found reviewer’s references highly relevant and included them in the manuscript.

5. Minor: Page 19: eq for normalized weights seems wrong (square root bar ends too soon..)

We updated this equation in the Methods section on page 20 for clarity.

Reviewer 3

1. I am wondering whether it’s possible to separate the “fixations on partners” into more detailed variables, such as “fixations on head” or “fixations on body” (relevant to Fig 2a). I am very curious to know, in the later session analysis, the transition probability corresponding to “view partner -> push” and “push -> view partner” are related to viewing different parts of the partner. Could it be “view partner’s head -> push”? Given that the authors used multi-point pose tracking, this can be tested and might provide more nuanced information on how they were using the social cues.

We appreciate reviewer’s interest in ‘zooming in’ on fixations on specific body parts of the partner monkey. Overall, there were more fixations on partner’s body than on the head. In the first monkey pair, there were 1,350 fixations on the body, compared to 836 fixations on the head (session average). In the second pair, nearly all fixations on the partner monkey were on the body, with some sessions having no fixations at all on the head. This is likely because eye contact can be aversive or a sign of threat in macaques (the second pair included two monkeys that were further separated in social hierarchy compared to those in the first pair). Therefore, we repeated the HMM analysis of social events in the first pair only, taking into account fixations on both the head and body of the partner. The results (Fig. R1) were similar to Fig. 2d and Extended Data Figure 2. Increasing transitional probabilities between pushing and viewing the partner are due to viewing the partner’s body.

Figure R1. Transitional Probabilities including viewing the head and body of the partner. Transitional probabilities from Markov Modeling estimation are plotted across sessions for each type of fixation on the partner monkey and each push event pair combination in monkey pair 1. The P value and fit from simple linear regression is included if $P < 0.05$.

2. It is unclear to me why the transition probability between “self push” and “partner push” was low, and did not increase with training? (relevant to Fig 2e). I would expect that this transition probability to also increase with training, because essentially it’s reflecting the cooperation (two animals pushing almost at the same time) to become more efficient at obtaining rewards. Maybe it’s due to the time bin size the authors used to create the behavioral event time series. I wonder how would this look if one uses a larger time bin that might fit better with a longer timescale. From Fig 1e, it seems that there was a 1-2 s gap between the two pushes, so maybe using a time bin that reflects this time gap might be more appropriate or at least informative.

The reviewer states that they expected the transitional probability between self-push and partner-push to be higher and increase during learning. However, this does not happen because fixation events usually occur in between pushes, and the analysis computes probabilities between consecutive events. The input to the model (*hmmestimate*, Matlab 2020b) is the sequence of states or events (pushes and ‘social cue’ fixations) that occur across a trial. All trials within a session that contained at least one occurrence of each event were included in analysis. The data is not binned. First, there is a sorted sequence of timestamps where each timestamp reflects an event that occurred. Then the timestamps are labeled accordingly: 1 = view reward, 2 = self push, 3 = view partner, and 4 = partner push. This results in a sequence of numbers which is the input to the model (‘behavioral event sequence’, below). The model computed transitional probabilities between two consecutive events, and it is rare that each animal’s push, events 2 and 4, occur

consecutively when fixations are considered. For instance, see below, we have included sequences that occurred across a few trials in one session (most sequences are much longer, but we selected short ones for visualization). As can be seen from these examples, a fixation on the reward or partner usually occurs between self and partner pushes.

Trial number	Behavioral event sequences
3	[1,2,1,3,4,4]
4	[1,2,3,4,4]
8	[3,1,2,4,4]
14	[2,3,3,3,1,4,4]
17	[2,1,3,3,3,3,4,4]
39	[3,4,2,1,4,4]
52	[1,1,2,3,4,4]
53	[1,2,1,3,3,4,4]
68	[1,2,3,4,4]
91	[2,3,2,1,4,4]

3. I was not sure why the number of green bars is smaller than the purple bars in the top panel of Fig. 3c (based on x axis values), but they seem to be higher in the bottom line-plot panel (based on the y axis values).

This is an excellent observation. As the reviewer states, the values for green bars in the top panel of Fig. 3C are smaller than those in the bottom panel because the top panel includes the percentage of pushes with preceding fixations only on the partner monkey, whereas the bottom panel includes preceding fixations on the partner monkey and/or reward system. Essentially, there are more self-pushes with preceding fixations on the reward and partner than just the partner alone. In contrast, partner pushes have more preceding fixations on the partner monkey than the reward. Indeed, according to Extended Data Fig. 4C, the number of reward fixations before self pushes is greater than the number of partner fixations before self pushes (and vice versa for partner pushes). Having said this, the reviewer’s question made us realize a minor error in the histograms for how we computed the percentage of partner pushes with fixations on the partner only (purple histograms, Fig. 3c top). That is, we previously calculated the percentage of pushes that had preceding fixations on the partner occurring at any time before the push. However, to be consistent with neural modulation during pushing and other analyses in the manuscript, we should only use the fixations that occurred 1000 ms around the push. We thus corrected the histograms in Fig. 3c-top where we only considered fixations on the partner during the correct time window. Overall, the original finding is unchanged (i.e., the self monkey still views the partner monkey significantly more during partner pushes compared to his own push) but the x-axis values for partner choice are now comparable with those in Fig 3c–bottom.

4. If I understand it correctly, the major difference between “fixation on social cue” and “fixation on partner” is that social cue also included reward.

4.1 Could the authors clarify what types of “view reward” was included as the social cue? Did it only include the rewarding event or the reward itself of the partner monkey?

We can certainly clarify how we categorized fixations. ‘View reward’ includes fixations on the food reward system: the pellet dispenser and tray (note that the tray always contained pellets during the fixations

on it. The only time a tray did not contain food reward is during the intertrial interval, and we did not include fixations from this period in analysis. ‘View partner’ includes all fixations on the partner monkey (head and body). ‘Social cues’ includes all the fixations categorized as ‘view reward’ and ‘view partner’ fixations. Non-fixation rewarding events, such as the start of the trial when pellets are dispensed and the end of the trial when pellets are received, were not included in any analyses. While viewing the partner might be subjectively rewarding, we cannot confirm this. Therefore, the ‘view reward’ category only includes fixations on the food reward. We updated the language in the “Viewing social cues drives cooperation during learning” on page 4 and Methods page 19 for clarification.

4.2 Going back to Fig. 3c, does the difference basically indicate that there were a lot more pushes with fixation on reward in “self choice” than in the “partner choice”?

Yes, the self-monkey viewed the reward more frequently before his own push than before the partner push (but he viewed the partner monkey more frequently before partner push than his own push). This was consistent across animal pairs (cf. Extended Data Fig. 4C).

5. Spike cross-correlation should be sensitive to the neurons’ firing rate (higher firing rates in at least one of the neuron of the pair relate to high cross-correlation), (5.1) did the authors address this confound in their analysis? I am wondering whether the higher spike correlations during the behavior events were largely due to the higher firing rate during the same time window.

Yes, our spike cross-correlation analysis uses two corrections to account for 1) stimulus induced firing rates/correlations and 2) changes in firing rates of individual neurons. We updated the Methods section, pages 20-21, to clarify this. Specifically, dividing by the geometric mean in Eq. 5 results in CCG peaks with relatively constant areas as firing rates of individual neurons change (Bair et al., 2001; Kruger and Aiple, 1988). We further corrected the CCG to remove any stimulus induced correlations by subtracting an all-way shuffle predictor – efficiently computed from the cross-correlation of the post-stimulus time histograms (PSTHs) (Bair et al., 2001; Pojoga et al., 2020; Shahidi et al., 2019).

5.2 To test the directionality of the communication between the brain areas, it’s common to analyze the spike-field coherence. Did the authors have tried analyzing the spike-field relationship or some directionality analysis using LFP (such as granger or PDC, or DTF)? Ideally, it should reveal the similar finding as presented here, and even add a new dimension (i.e., frequency).

We did not analyze the LFP data that was recorded wirelessly during the experiments. The reason was the high-quality and abundant spiking data from 2 cortical areas acquired across sessions and animals. While we agree that the methodology suggested by the reviewer might reveal some insight, it would nonetheless not allow us to examine how spike timing coordination contributes to the encoding of social events (Figure 5E). We believe that the directionality of the communication between cortical areas would be best examined more directly by employing optogenetics studies similar to those performed by us in Debes and Dragoi, 2023 (in relation to cortical feedback). Feedforward and feedback interactions during social interactions constitute an ongoing research interest in the lab, and our future studies will address the issues raised by the reviewer.

6. About Fig. 5d, there seems to be in general higher number of “V4 leads” than “dlPFC leads” but there are a plenty of cells showing the opposite pattern as well (the differences are not as one-sided) and thus it’s unclear if these proportion differences are significant.

Reviewer’s observation is correct. We explored this further and performed a paired signed-rank test between the percentage of cell pairs with positive and negative time lags within each session (Fig. R2).

We found that for all events except partner choice, the percentage of correlated pairs with positive time lag (V4 leads) was not significantly greater than the percentage with negative time lags (dIPFC leads). Therefore, we edited the manuscript text in “Spike-timing coordination increases during learning cooperation” section, page 10, to reflect this result. Additional experiments involving causal manipulations are likely necessary to determine directionality of information between V4 and dIPFC.

Figure R2. Percentage of correlated cell pairs with different time lags. For each session (open circles), the percentage of cell pairs with maximum coincident spiking at positive and negative time lags from CCG analysis during each social event is plotted. Plots include all sessions from each monkey. The red cross represents the session-average percentage of pairs. The P-value from Wilcoxon signed-rank test comparing positive and negative lag distributions is displayed for each social event.

7. One of the concerns that drew a lot of my attention is that there’s no real non-social control in this task. Ideally, if one wants to claim that animals were doing the task in a social and cooperative manner, it’s better to do controls such as blocking the view of seeing the other animal, two animals pushing without cooperative contingency, or fake or computer partner instead of a real monkey. However, after reading this manuscript, I am generally convinced that the two animals WERE cooperating. I believe that the learning paradigm really helped here – animals had an increased level of coincident pushes over training, and the transition probability between social events and push also increased.

7.1 It could be better if the authors also had a control showing that if there is no need to cooperate, there will be no social view to push transition. But I understand that the authors did not particularly claim anything about social specificity of the findings – in fact, the findings themselves are very interesting still given the naturalistic paradigm and environmental contingencies involving another conspecific. Also, I can see that it would be hard to have such controls, especially the authors observed and reported the learning phase – e.g., once the animals learned, it’s hard to have a pure control because they cannot unlearn what they learned and they may still use the social strategy to do the non-social controls. Regardless, I would like to see the authors’ comments about the missing controls in Discussion to guide the readers on the interpretation. Just to be clear, I am NOT recommending adding any non-social control here.

We thank the reviewer for these remarks and suggestions. There were two non-social controls that we performed one month after the end of the learning sessions: 1) animals completed the cooperation task with an opaque divider between them, and 2) social-and-solo trial sessions (originally included in the manuscript, Extended Data Figure 8b-d) which contained blocks of social and solo trials where the partner monkey was removed from the arena and self-monkey obtained reward by pushing his own button (for explanations of both controls, see Methods - *Behavioral training and experiments*). We updated page 4 of the manuscript to include outcomes from the opaque divider control, included the results in Extended Data Figure 8a (below), and added the experimental paradigm to the Methods section. The purpose of the social-and-solo control experiment was to explore whether pre-push activity of the self-monkey represents a specifically social decision to cooperate. Analysis of pre-push activity across social and solo conditions revealed social context influences self-choice, and neural representations for self-pushing are significantly different in solo and social conditions (Extended Data Fig. 8d). We updated our discussion of these findings on page 8 of the manuscript.

Extended Data Fig. 8 | Non-social controls. a, Left - average amount of time between self and partner monkey presses during learning (‘with viewing’) sessions and control sessions with the opaque divider (‘without viewing’). $P = 2.30e-08$, Wilcoxon rank sum test. **Right** - average delay to cooperate, or time for both monkeys to be pressing from the start of a trial, during learning sessions and control sessions with the opaque divider. $P = 4.64e-08$, Wilcoxon rank sum test. Times were pooled across sessions ($n = 4$ sessions for each condition) and averaged across monkeys. SEM is represented with error bars. * $P < 0.05$, ** $P < 0.01$, *** $P < 0.001$.

7.2 In addition, it is not too surprising that dlPFC and V4 neurons generally did not show much agent specificity given that these areas prioritize sensory and motor aspects of complex behaviors (as opposed to, say, medial areas). For example, Fig. 4f being a “social learning model” or more generic “learning model” in a complex environment remains unclear.

We agree with the reviewer that our cells did not show agent specificity but rather mixed selectivity to social events, particularly in dlPFC (Fig. 3e). Additionally, the reviewer asks if our “social learning model” in Fig. 5f is only a model for social learning. Since our learning task only included social behavior, we are currently unable to determine if the model can generalize. However, given prior research studies of learning in other contexts, it may be possible that our model could apply to other forms of learning in naturalistic environments.

8. The authors provided evidence that neural selectivity changes over the course of learning. The interpretation is that the coding scheme is shifting more toward a distributed coding scheme.

8.1 However, it is less clear if the coding is becoming more distributed per se or the coding is being degraded over time, in a sense of representational decay, or changing over learning, in a sense that the information is more allocated toward some downstream areas.

The representation of social variables does not appear to become degraded over time or allocated to a downstream area as evidenced by improved decoding performance in each area during learning (Fig. 4). Additionally, we chronically and stably recorded from the same neural population across learning sessions (> 70% stable units, Extended Data Fig. 3) within a four-month period. We used stringent and established measures based on waveform shape to determine population stability (see Methods, *Identifying stable units across sessions*). Moreover, neural analyses using only the stable units yielded the same results as the entire population (Extended Data Fig. 12). Finally, for each social event, we did not find any systematic changes in neural firing rate, response latency, or D-prime values across sessions (Extended Data Fig. 9 and 11). While some (~15%) individual neurons' D-prime values can increase or decrease across sessions (Extended Data Fig. 9d), this does not reflect a decay in representation but rather a change in the preferred event or a developed preference during learning.

8.2 Do authors propose the distributed coding scheme based on the decrease in weight distribution specifically for social cues where the decoding performance increased?

Yes, the decrease in variance, kurtosis, skewedness or maximum value of neural population weight distributions only occurred in models where decoding performance increased across sessions. Extended Data Figure 9b summarizes this finding across decoding models (social cues, categories, choice, and non-social cues).

8.3 I wonder if this specific combination of results could also happen when neurons begin to be more and more de-correlated to signal one (or just few) specific events in the behavioral environment over the course of the session (as opposed to more and more neurons are beginning to represent the shared information as in a distributed coding scheme)?

Based on the individual neuron D-prime analysis (Extended Data Fig. 9c-d), only few neurons appeared to exhibit changes in selectivity or develop a preference to one social event over the other during learning. However, we did observe changes in the population level during learning, both in decoding performance and weight distribution. It is possible the changes of these few neurons mediate some of the ensemble changes we measured, but we are unable to know for certain the underlying causes of the improvement in population coding during learning. We interpret the decrease in variance of weight distributions to mean that early during learning, a select number of cells contribute most to social interactions relative to the rest of the population. However, while learning progressed, the magnitude of the weights decreased and information about social events became distributed more evenly across the population.

Regarding the correlation question, we assume the reviewer is referring to the type of coding scheme employed during social interactions and learning. However, this could be a large project in itself and we are conducting it in parallel in our lab. Briefly, we found significant sparseness of the visual/social environment, both in V4 and dlPFC (more sparseness in prefrontal cortex), that is significantly related to the ability of the neural population to discriminate information. Ongoing population analyses are currently employed to examine the changes in coding scheme during the time course of learning. Our preliminary analyses though indicate no systematic/significant trend of population sparseness in either cortical area during the time course of learning.

Reviewer Reports on the First Revision:

Referees' comments:

Referee #1 (Remarks to the Author):

The authors took our concerns seriously and addressed them all satisfactorily. The claims made in the paper seem all solid and well substantiated and reasoned.

Referee #3 (Remarks to the Author):

The authors have addressed my comments through additional data analyses and clarifications. I have no further comment, and I hope this work opens up exciting work in the future.

--

Signed below to opt in for Nature's new transparent peer review scheme: Reviewed by Steve W. C. Chang with assistance from Weikang Shi, a postdoctoral fellow in his lab.